# A Particle Filter scheme for multivariate data assimilation into a point-scale snowpack model in Alpine environment

Gaia Piazzi[1], Guillaume Thirel[2], Lorenzo Campo[1], Simone Gabellani[1]

[1]CIMA Research Foundation, Savona, 17100, Italy

[2] Catchment Hydrology Research Group, HYCAR Research Unit, Irstea, Antony, 92160, France

*Correspondence to*: Gaia Piazzi (gaia.piazzi@cimafoundation.org)

**Abstract.** The accuracy of hydrological predictions in snow-dominated regions deeply depends on the quality of the snowpack simulations, whose dynamics strongly affects the local hydrological regime, especially during the melting period. With the aim of reducing the modelling uncertainty, data assimilation techniques are increasingly being implemented for operational purposes. This study aims at investigating the performance of a multivariate Sequential Importance Resampling - Particle Filter scheme designed to jointly assimilate several ground-based snow observations. The system, which relies on a multilayer energy-balance snow model, has been tested at three Alpine sites: Col de Porte (France), Torgnon (Italy), and Weissfluhjoch (Switzerland). The implementation of a multivariate data assimilation scheme faces several challenging issues, which are here addressed and extensively discussed: (1) the effectiveness of the perturbation of the meteorological forcing data in preventing the sample impoverishment; (2) the impact of the parameters perturbation on the filter updating of the snowpack state; the system sensitivity to (3) the frequency of the assimilated observations and (4) the ensemble size. The perturbation of the meteorological forcing data turns out to be generally not sufficient to prevent the sample impoverishment of the particles sample, which is highly limited when jointly perturbating key model parameters. The parameters perturbation proves, however, to sharpen the system sensitivity to the frequency of the assimilated observations, which can be successfully relaxed by introducing indirectly estimated information on snow mass-related variables. The ensemble size is found not to greatly impact the filter performance in this point-scale application.

## 1. Introduction

Snow-dominated areas play a distinctive role in water supply in terms of soil moisture, runoff, and groundwater recharge (Vivoroli et al., 2007; Dettinger, 2014). The knowledge of the spatio-temporal distribution of snow cover is therefore of critical importance to several applications (Viviroli et al., 2011; Fayad et al., 2017). When dealing with hydrological predictions in mountain regions, the modelling of snow dynamics is a challenging issue due to complex interactions among site-dependent factors, namely the meteorological forcing (Bormann et al., 2013; Luce et al., 2014), local topography (Molotch and Meromy, 2014; Revuelto et al., 2014), the presence of vegetation and the wind-induced phenomena (Gascoin et al., 2013; Zheng et al., 2016; Quéno et al., 2016).

Recently, an increasing interest focuses on investigating the potentials of Data Assimilation (DA) schemes in consistently improving the model simulations by assimilating ground-based measurements or remotely sensed snow-related observations (Bergeron et al., 2016; Dziubanski and Franz, 2016; Griessinger et al., 2016; Huang et al., 2017).

Several DA methodologies have been developed, each one characterized by different performances mainly according to its degree of complexity. The sequential DA techniques are widely used for real-time applications, since they allow taking benefit from the observational data as it becomes available and sequentially update the model state. The most basic approach relies on the direct insertion (Liston et al., 1999; Rodell and Houser, 2004; Malik et al., 2012), which promotes the simple replacement of model predictions with observations, whenever available. Even though this conceptually simple DA scheme is an attractive method, its implementation within complex, multi-layered snow models is not straightforward, mainly because of possible model shocks resulting from physical inconsistencies among state variables (Magnusson et al., 2017). More advanced are the optimal interpolation schemes (Brasnett,1999; Liston and Hiemstra, 2008), the Cressman scheme (Cressman, 1959; Drusch et al., 2004; Dee et al., 2011; Balsamo et al. 2015) and the nudging method (Stauffer and Seaman, 1990; Boni et al. 2010), allowing to take into account the observational uncertainty, which is a priori defined. At a higher level are the Kalman filters, which are among the most commonly used sequential DA techniques (Kalman, 1960). The standard version of the Kalman Filter (KF) (Gelb, 1974), which relies on the system linearity assumption, was upgraded to the Extended Kalman Filter (EKF) (Miller et al., 1994) allowing to deal with nonlinear dynamic models through a linearized statistical approach (Sun et al., 2004; Dong et al., 2007). With the aim of overcoming the need for a statistical linearization, which can be unfeasible when dealing with strongly nonlinear models, the Ensemble Kalman Filter (EnKF) has been developed (Evensen, 1994). Unlike the KF and EKF schemes, this method does not require a model linearization since the error estimates are evaluated from an ensemble of possible model realizations, commonly generated through the Monte Carlo approach (Evensen, 2003). In the recent past, an increasing number of studies on snow hydrology have contributed to confirm the EnKF as a well-performing technique enabling to enhance the accuracy of hydrological simulations by consistently updating model predictions through the assimilation of snow-related observations (Andreadis and Lettenmaier, 2005; Durand and Margulis, 2006; Clark et al., 2006; Slater and Clark, 2006; Su et al., 2008; Durand and Margulis, 2008; Su et al., 2010; De Lannoy et al., 2012; Magnusson et al., 2014; Griessinger et al., 2016; Huang et al., 2017).

Even though the EnKF scheme provides a flexible framework to explicitly handle both observational and modelling uncertainties (Salamon and Feyen, 2009), some constraining assumptions hinder filter performance (Chen, 2003). Firstly, in Kalman filtering the analysis step relies on the second-order moments (Moradkhani et al., 2005). However, because the state variables in stochastic-dynamic systems are modelled as random variables, the involved probability distributions are not supposed to follow a Gaussian distribution (Weerts and El Serafy, 2006). Thus, in strongly nonlinear systems the first two moments are not likely to be sufficient to properly approximate the posterior probability distributions, whose estimates require the tracking of higher-order moments (Moradkhani et al., 2005). Secondly, the EnKF is limited to the linear updating procedure with significant simplification affecting the filter performance. Recently, Piazzi et al. (2018) investigated the main limitations in implementing a multivariate EnKF scheme to assimilate ground-based and remotely-sensed snow data in the

framework of snow modelling. Furthermore, since the EnKF involves state-averaging operations, the implementation of this DA technique into highly-detailed complex snowpack models (e.g. with varying number of snow layers) is even more challenging, or even unfeasible.

In order to overcome these limiting issues, filter methods for non-Gaussian, nonlinear dynamical models have been developed. These sequential Monte Carlo techniques, also known as Particle Filter (PF) schemes (Gordon et al., 1993), have the main advantage of relaxing the need for restrictive assumptions on the form of the probability distributions, since the full prior density derived from the ensemble is used within the updating procedure (Arulampalam et al., 2002). Thanks to their suitability to better succeed in handling systems nonlinearities, PF schemes are currently garnering a growing attention for snow modelling applications. Leisenring and Moradkhani (2011) compared the performances of common sequential EnKF-based DA methods and PF variants at assimilating synthetic SWE measurements to improve its seasonal predictions and to estimate some sensitive parameters in a small-scale snowpack model. The results suggested that all the DA techniques succeeded in enhancing the SWE dynamics. Even though PF-based techniques generally revealed a higher accuracy, the resulting bias was comparable with the Kalman filters one. Dechant and Moradkhani (2011) evaluated the PF performance in assimilating remotely-sensed microwave radiance data to update the states of a snow model. The results showed that the DA scheme allowed to improve simulations of SWE as well as discharge forecasts. Thirel et al. (2013) investigated the implementation of the PF technique to assimilate MODIS SCA data into a physical distributed hydrological model, in order to enhance snowmelt-related stream flow predictions. Both synthetic and real experiments showed clear improvements of model discharge simulations, especially for intermediate values of observation error. Margulis et al. (2015) tested a newly-proposed PF approach to improve SWE estimates when assimilating historical Landsat-derived observations of fractional snow-covered area into a land surface model. This technique has been recently applied by Cortés et al. (2016). Charrois et al. (2016) investigated the performances of the Sequential Importance Resampling PF (SIR-PF) scheme in assimilating MODIS-like synthetic data of optical reflectance into a detailed multilayer snowpack model. The study assessed the impact of the assimilation, which well succeeded in reducing RMSE values on both snow depth and SWE with a resulting reduction of the uncertainty on the snow melt-out date. An even larger bias reduction was achieved by updating the model assimilating synthetic snow depth observations, except for thin snowpack. However, they proved that the joint assimilation of remotely sensed reflectance and measurements of snow depth can be the best combination to provide a significant improvement of the model simulations at the local scale. Magnusson et al. (2017) found that the assimilation of daily snow depth measurements within a multi-layer energy-balance snow model through the PF scheme resulted in an improvement of the simulations of SWE and snowpack runoff over the whole analysis period. However, model daily runoff dynamics did not substantially benefit from the snow depths assimilation, except during the melt-out period.

In view of the promising performances of PF-based schemes in snow-related univariate DA applications, this study aims at contributing to this research field by investigating the potential of this technique in performing multivariate DA. Main issues of key importance are therefore addressed: (1) How does the PF scheme succeed in consistently updating the snowpack system state by jointly assimilating several in-situ snow-related point data? (2) What are the most constraining limitations in

implementing a multivariate PF-based scheme in the framework of snow modelling and how to overcome them? (3) What is the impact of the uncertainties of meteorological data and model parameters on the filter effectiveness? (4) How much does the filter performance depend on the observations availability? What is an effective approach to limit the system sensitivity to the difference in the measurement frequency of the assimilated variables?

Section 2 firstly describes the analysed case studies and the modelling system consisting of a multilayer energy-balance model and the DA scheme, whose main features are discussed. After sketching the experimental design, Section 3 presents and assesses the main results of the experiments on different configurations of the multivariate DA scheme. The main issues hindering the filter efficiency are thoroughly discussed by analysing the impact of the meteorological perturbation, the uncertainty of model parameters and the use of an additional snow density model to reduce the system sensitivity to the in-situ measurement frequency. Lastly, conclusions are outlined in Section 4.

## 2. Materials and methods

### 2.1 Case studies

With the aim of testing the snow modelling system at a point-scale, the selection of the case studies has been restricted among pilot experimental sites, where automated weather stations supply meteorological and snow-related measurements of high quality and completeness. This choice has allowed to reliably investigate the filter performance regardless of possible inconsistent measures, since generally the in-situ observations are extensively verified through a quality control and data gaps filling (Morin et al., 2012; Essery et al., 2013; Lafaysse et al., 2017). With the purpose of investigating the snow model sensitivity to various meteorological conditions, measurement sites located at different elevations have been chosen. Moreover, the selection has been limited over the domain of interest, namely the Alpine region. Among the Alpine measurement sites, three snow experimental sites, meeting all the requirements to force and evaluate a snow model, have been selected: Col de Porte (France), Weissfluhjoch (Switzerland) and Torgnon (Italy).

_Col de Porte site_

The Col de Porte observatory (CDP) is located near Grenoble, in the Chartreuse massif in the French Alps (45°30' N, 5°77' E) at an elevation of 1325 m a.s.l.. This pilot site is placed in a grassy meadow surrounded by a coniferous forest on the eastern side. Snow cover is usually present from December to April, on average (Lafaysse et al., 2017). Nevertheless, during the winter season surface melt and rainfall events can frequently occur at the relatively low altitude of the experimental site. The mean annual precipitation is about 1110 mm of rain and 570 mm of solid precipitation, and the air temperature falls below 0°C generally only during December and March. In-situ meteorological data, at the hourly resolution, include measurements of 2-m air temperature and relative humidity, 10-m wind speed, incoming short- and longwave radiations and precipitation rates. Precipitation phase is manually assessed using all possible ancillary information (Lafaysse et al., 2017). Snow-related observations are provided both at daily and hourly resolution. Along with weekly manual SWE measurements, since the season 2001-2002 SWE is automatically measured on a daily basis by a ground-based cosmic rays counter. Hourly

snow albedo data are estimated through the radiation sensors, as the ratio between incoming and reflected shortwave radiation (Morin et al., 2012). Moreover, measurements of snow surface temperature are hourly available.

*Weissfluhjoch site*

The Weissfluhjoch site (WFJ) (46.82°N, 9.80°E) is located at an altitude of 2540 m a.s.l. in the Swiss Alps, near Davos, Switzerland (WSL Institute for Snow and Avalanche Research SLF, 2015b). This snow experimental site is placed in an almost flat area of a south-easterly oriented slope. At WFJ the snow season generally starts in October/November and lasts until June/July (Wever et al., 2015). The average air temperature exceeds 0°C generally only between May and October and the mean annual precipitation is about 1180 mm of solid precipitation and 556 mm of rain. An automated weather station provides a comprehensive multiyear dataset including measurements of air temperature and relative humidity, wind speed and direction, incoming and outgoing short- and longwave radiation, snow/ice surface temperature, snow depth and precipitation (Schmucki et al., 2014). The rain gauge measures at an interval of 10 min, whereas most other measurements are done at 30-min intervals. Every two weeks, generally in early and in the middle of each month depending on weather conditions, a manual full-depth snow profile is performed in order to provide measurements of snow temperature and snow density (WSL Institute for Snow and Avalanche Research SLF, 2015a). Snow density and SWE are also manually measured through snow cores.

*Torgnon site*

The Torgnon site (TGN) (Tellinod, 45°50' N, 7°34' E) is located in Aosta Valley, a mountain region in north-western Italian Alps. The experimental site is subalpine grassland, at an elevation of 2160 m a.s.l. (Filippa et al., 2015). The area slopes lightly (4°) and it is characterized by a typical intra-alpine semi-continental climate, with an average annual temperature of around 3°C and a mean annual precipitation of 880 mm (Galvagno et al., 2013). On average, the snow season lasts from the end of October to late May, when the test site is covered by a thick snow cover (90-120 cm). Since 2008, an automatic weather station provides 30-min averaged records of different meteorological parameters, including air and surface temperatures, incoming and outgoing short- and longwave radiations, and surface albedo, precipitation, soil water content, snow depth and wind speed and direction. Furthermore, SWE in-situ measurements are available with a sampling resolution of 6 hours for the snow seasons 2013-2014 and 2015-2016. The sensor provides comprehensive SWE measures over a sizeable area of 50-100 m$^2$, with a resulting lower impact of several local factors (e.g. snow drifting, vegetation). Bi-weekly manual measures of snow density (snow pits) are available during the winter season (3-4 measurements per month, on average).

According to the in-situ meteorological observations, the selected experimental sites are characterized through two of the most ruling climate forcing, namely air temperature and snowfall rate (Lòpez-Moreno and Nogués-Bravo, 2005), together with the snow depth trend (Figure 1).

## 2.2 Snow model

The snow model relies on a multilayer scheme consisting of two layers of both soil and snowpack. The model provides an estimate of several snow-related variables describing the snowpack state by simulating the main physical processes (i.e. accumulation, density dynamics, melting and sublimation processes, radiative balance, heat and mass exchanges). The explicit energy and mass balances framework requires several input forcing meteorological data: air temperature, wind velocity, relative air humidity, precipitation and incident shortwave solar radiation. It is noteworthy that the incoming longwave radiation is estimated through the Stephan-Boltzmann law, as a function of air temperature and time-variant air emissivity. While a full description of model is extensively explained in Piazzi et al. (2018), some details on model parameterizations are given below.

- *Snowpack layering*: The mass transfer between the two snow layers is ruled by an empirical parameterization allowing to maintain the surface layer thinner than the underlying one, with the aim of properly approximating the surface temperature.

- *Precipitation phase*: When total precipitation rate is provided, the partitioning between rain- and snowfall is based on both air temperature and relative humidity, according to the approach proposed by Froidurot et al. (2014).

- *Snow compaction*: Snow density is updated considering both the compaction and the destructive thermal metamorphism according to the physically-based parameterization proposed by Anderson (1976).

- *Fresh snow density:* In case of snowfall, the fresh snow density is evaluated as a function of the air temperature (Hedstromand and Pomeroy, 1998).

- *Snow albedo:* With respect to the original version of the model scheme described in Piazzi et al. (2018), the empirical snow albedo parameterization proposed by Douville et al. (1995) has been introduced. Following this formulation, the surface albedo ($\alpha$) is predicted as prognostic variable:

Albedo for cold snow: $\alpha_s(t + \delta t) = \alpha_s(t) - \tau_\alpha^{-1}\delta t$         (1a)

Albedo for melting snow: $\alpha_s(t + \delta t) = [\alpha_s(t) - \alpha_{min}]exp(-\tau_m^{-1}\delta t) + \alpha_{min}$     (1b)

Albedo update (snowfall event): $\delta\alpha_s = (\alpha_{max} - \alpha_s)\frac{S_f\delta t}{S_0}$       (1c)

where:

$S_f$ is the snowfall rate [kg m$^{-2}$]; $\alpha_{max}$= 0.85; $\alpha_{min}$= 0.5; $S_0$ = 10 kg m$^{-2}$; $\tau_\alpha$ = $10^7$ s; $\tau_m$ = 3.6 · $10^5$ s.

The albedo dynamics is described by a linear decay over time under cold snow conditions (Eq. 1a) and an exponential decay in the presence of melting snow (Eq. 1b). When a snowfall event occurs, the albedo is consistently updated (Eq. 1c).

- *Turbulent heat fluxes:* Sensible and latent heat fluxes are evaluated following the bulk formulation. The atmospheric stability is evaluated as a function of the Richardson Bulk number, according to the empirical scheme of Caparrini et al. (2004).

## 2.3 Particle filter data assimilation scheme

The PF technique relies on the Monte Carlo approach to solve the Bayesian recursive estimation problem. Consider the state vector ($X_t$) including all the prognostic variables:

$$X_t = M[X_{t-1}, \theta, U_t, \Omega_t] \tag{2}$$

where $M$ is the dynamic model operator, which calls for the meteorological input vector ($U_t$), the vector of model parameters ($\theta$), and the model error ($\Omega_t$). Whenever a set of observations is available, the analysis procedure allows to update the a priori state according to the observation vector ($Y_t$), which requires an observation operator ($H$) enabling to generate the model equivalents of the observations:

$$Y_t = H[X_t, \Psi_t] \tag{3}$$

where $\Psi_t$ is the observation error, which is generally assumed to be Gaussian and independent of the model error.

The sequential filtering problem aims at finding the maximum of the conditional probability density function (pdf) of the model state $P(X_t|D_t)$, where $D_t = \{Y_t; t = 1, \dots, t\}$ encompasses all the available observational information on the time step $t$.

Given the posterior pdf at time $t$-$1$ $p(X_{t-1}|D_{t-1})$, it is possible to obtain the pdf of the current state $p(X_t|D_t)$ in two stages, namely the prediction of the prior density $p(X_t|D_{t-1})$ (Eq. 4) and the update of the forecast pdf according to new observations (Eq. 5).

$$p(X_t|D_{t-1}) = \int p(X_t|X_{t-1})p(X_{t-1}|D_{t-1})\,dX_{t-1} \tag{4}$$

$$p(X_t|D_t) = p(X_t|Y_t, D_{t-1}) = \frac{p(Y_t|X_t)p(X_t|D_{t-1})}{\int p(Y_t|X_t)p(X_t|D_{t-1})dX_t} \tag{5}$$

where $p(X_t|X_{t-1})$ is the known transition pdf, $p(Y_t|X_t)$ measures the likelihood of a given model state with respect to the observations.

When dealing with high-dimensional and nonlinear systems, an analytical solution of the problem is unfeasible (Moradkhani et al., 2005). The implementation of ensemble methods (e.g. Monte Carlo sampling) allows to fully approximate the posterior density $p(X_t|D_t)$ through a set of $N$ independent randomly drawn samples, called particles (Arulampalam et al., 2002; Moradkhani et al. 2005; Weerts and El Serafy, 2006):

$$p(X_{0:t}|Y_{1:t}) \approx \sum_{i=1}^{N} W_t^i \delta(X_{0:t} - X_{0:t}^i) \tag{6}$$

where $\{X_t^i, W_t^i\}$ denote the $i$-th particle drawn from the posterior distribution and its associated weight, $\delta(\cdot)$ is the Dirac delta function. It is noteworthy to consider that the direct sampling of particles from the posterior density is generally difficult, since its distribution is often non-Gaussian. Therefore, particles ($X_t^i$) are drawn from a known proposal distribution $q(X_{0:t}^i|Y_{1:t})$, according to the Sequential Importance Sampling (SIS) approach (Moradkhani et al., 2005; Guingla et al., 2012). The importance weights of the particles are recursively defined according to the following formula:

$$W_t^i \propto W_{t-1}^i p(Y_t|X_t^i) \tag{7}$$

A well-known common issue with SIS-PF is the sample degeneracy, which prevents particles from properly approximating the posterior distribution. Arulampalam et al., (2002) explained that whenever the effective number of particles ($N_{eff}$) falls below a fixed threshold value, the impact of the degeneracy needs to be mitigated by increasing the number of particles, where:

$\quad N_{eff} \approx \dfrac{1}{\sum_{i=1}^{N}(W_t^i)^2}$ (8)

Since this approach is often unfeasible due to the increase in computational demand (Salamon and Feyen, 2009), a resampling procedure is frequently introduced to restore the sample variety through a Markov chain Monte Carlo (Moradkhani et al., 2005).

### 2.3.1 Sequential Importance Resampling

Gordon et al. (1993) proposed the Sequential Importance Resampling (SIR) technique, which introduces a resampling procedure within the SIS procedure. At each time step, the additional resampling step discards particles having low importance weights while replicating particles having high importance weight, while the total number of particles $N$ is maintained unchanged (Figure 2a, b). As exhaustively explained by Weerts and Serafy (2006), the SIR algorithm relies on the generation of an empirical cumulative distribution (cdf) of the particles according to their weights $W_t^i$ (Figure 2c) and the

projection of a discrete set of $N$ samples $\{X_t^i, i = 1, \dots, N\}$ with probabilities $\{W_t^i, i = 1, \dots, N\}$ uniformly drawn within the domain of the distribution. The resulting set contains replications of the particles having high importance weight, which are the most likely to be drawn (Figure 2d).

### 2.3.2 Likelihood function

When dealing with a multivariate SIR-PF scheme, it is necessary to take into account the different uncertainties affecting

each observed variable. Therefore, the likelihood function is a $N_{obs}$-dimensional normal distribution, where $N_{obs}$ is the varying number of the effectively assimilated variables. The likelihood function is therefore defined as:

$p(Y_t|X_t) = N\{(Y_t - X_t^i), \mu, R\}$ (9)

where $\mu$ and $R$ are respectively the null mean vector and the error covariance matrix of observations characterizing the multivariate Gaussian distribution. Thus, at each assimilation time step the particles weights are updated according to the

following equation:

$W_t^i = \dfrac{exp\left(-\frac{1}{2R}\left[Y_t - H\left(X_t^i\right)\right]^2\right)}{\sum_{i=1}^{N} exp\left(-\frac{1}{2R}\left[Y_t - H\left(X_t^i\right)\right]^2\right)}$ (10)

The importance weight of each particle is updated according to its likelihood value depending on how it is placed with respect to all the available observations.

**2.4 Generation of ensemble particles**

**2.4.1 Perturbation of meteorological input data**

Meteorological forcings are one of the major sources of uncertainty affecting snowpack simulations (Raleigh et al., 2015). Therefore, an ensemble of possible model realizations is generated by perturbing the model inputs, namely precipitation intensity, air temperature and relative humidity, solar radiation, wind speed. The ensemble of perturbed inputs allows to take into account a well-representative range of weather conditions at the experimental sites, which result in an ensemble of possible snowpack states standing for the uncertainty of model predictions (Charrois et al., 2016). A meteorological ensemble of 100 members is generated by perturbing the in-situ meteorological data with an additive stochastic noise applied (in a log-scale for precipitation and wind speed) at each time step (i.e. 15 minutes). Following the methodology proposed by Magnusson et al. (2017), the random perturbations are provided through a first-order autoregressive model in order to guarantee a physical consistency and a temporal correlation of the time-variant forcings. Perturbations are generated considering the error statistics evaluated at the CDP site (Table 1) (Magnusson et al., 2017), which result from the comparison between SAFRAN reanalysis data (Vidal et al., 2010) and the observations supplied by the French station (Charrois et al., 2016). Even though this approach ensures to take account of the actual meteorological errors affecting the quality of the model predictions, the main limitation of this procedure is the lack of correlations among the perturbed forcing variables, which does not ensure their physical consistency (Charrois et al., 2016).

It is also noteworthy that the same statistics of the meteorological analysis error specifically derived at the CDP station are used for the generation of the meteorological ensembles at all the snow experimental sites. As highlighted by Magnusson et al. (2017), this approach is likely to reduce the filter performance at the Italian and Swiss sites.

**2.4.2 Perturbation of model parameters**

Alongside the meteorological forcing, the parameterization of the physical processes occurring within the snowpack contributes to greatly increasing the uncertainty affecting model predictions (Essery et al., 2013; Lafaysse et al., 2017). The perturbation of key model parameters allows to take account of the uncertainty resulting from their empirical estimation. Furthermore, since the introduction of stochastic noise plays a major role in reducing the effect of the sample impoverishment (Moradkhani et al., 2005), the uncertainty of model parameters is supposed to contribute to restore the ensembles spread between two following assimilation time steps. Following the methodology proposed by Moradkhani et al. (2005), the resampling procedure is carried out both in the parameters and the state variables spaces. Therefore, at each assimilation time step, after the particles resampling the parameters are perturbed through an additive noise before being used at the successive time step. Following Salamon and Feyen (2009), the parameters variance is restricted between upper and lower limits in order to avoid model instabilities and to also assure a minimum process noise, in order to prevent any variance collapse. The variance ranges are set according to the results of several tests carried out by varying their limits and evaluating the impact on filter performance. Unlike the study of Moradkhani et al. (2005), who applied the dual SIR-PF

scheme to estimate model parameters, in this case the main aim is to succeed in enlarging the parameters ensemble through a consistent perturbation variance to ensure a significant spread of the particles.

When considering the uncertainty of model parameters, a preliminary sensitivity analysis is of key importance for a twofold reason. Firstly, it is intended to properly identify the parameters mostly affecting the model simulations. Secondly, an accurate selection accordingly enables to neglect those parameters whose perturbation would demand for a larger computational requirement without resulting in a significant improvement of the ensemble spread.

Consistently with the study of Piazzi et al. (2018), snow roughness and snow viscosity are assumed to critically condition the model snowpack dynamics (Figure 3). Indeed, several analyses revealed a high sensitivity of model simulations to the perturbation of these mass-related parameters, especially in terms of SWE and snow density. Snow roughness strongly affects the snowpack energy balance by ruling the turbulent heat fluxes. As a consequence, the perturbation of this parameter mainly impacts the SWE ensembles by providing each particle with different snow melting fluxes. The effects of the perturbation of snow viscosity are prominent on the snow density evolution, especially on the snow compaction dynamics. As shown in Figure 3, it is noteworthy to observe the gradual increase of the ensemble spread of snow viscosity values throughout the melting period, suggesting that the perturbation of this parameter allows an offsetting effect of model melting issues.

Since the main criterion for selecting the model parameters focuses on identifying those whose perturbation allows to increase the ensemble spread of the state variables only slightly affected by the meteorological uncertainty, the three parameters describing the dynamics of the surface albedo are also considered (Table 2). The perturbation of the albedo parameters mainly guarantees a significant enlargement of the ensembles spread of this prognostic variable, with an impact on the snow mass balance especially during the melting period.

## 2.5 Experimental setup

### 2.5.1 Snow data

The multivariate DA scheme has been designed to consistently update the system state by jointly assimilating ground-based observations of surface temperature, albedo, snow depth, SWE and snow density. Table 3 lists the datasets of the experimental sites.

Automatic in-situ measurements of surface temperature, albedo, and snow depth are supplied on an hourly or sub-hourly basis by all the selected stations throughout the whole datasets.

Even though direct SWE measurements are generally widely lacking, the snow experimental sites are one of the main sources of consistent measures of this variable. Daily automatic measures are provided at CDP since the winter season 2001/02. With a lower measurement frequency, at WFJ SWE observations are available every two weeks over the whole dataset period. Unlike these two sites, the TGN station supplies in-situ 6-hours automatic SWE measurements during the snow seasons 2013/14 and 2015/16. In order to be able to properly use these measures within the analysis of simulations on

seasonal/annual scale, the raw observational data have been smoothed from possible inconsistent oscillations and anomalies (e.g. rain-on-snow events) through their daily average (Table 4).

Although an exhaustive knowledge of snow density is needed to properly define the snowpack state and its dynamics, direct continuous observations of snow density are generally lacking. An exception is the TGN site, where biweekly manual

measurements provide useful information throughout the whole dataset period. However, thanks to the relation among snow depth, SWE and snow density (Jonas et al., 2009), observations of at least two of these variables are enough to indirectly estimate the third one with a roughly tolerable degree of uncertainty. According to this approach, at both the Swiss and the French sites bi-weekly and daily snow density measurements have been derived, respectively (threshold value at 550 kg/m$^3$). At TGN daily snow densities have been indirectly estimated during the two winter seasons when SWE measures are

available. Conversely, bi-weekly SWE estimates have been derived during the other two snow seasons, when snow densities measurements are otherwise supplied. Because the two sensors measuring snow depth and SWE can be not located at exactly the same point, however, it is noteworthy that possible inconsistencies can arise due to the spatial variability in snow cover, especially under shallow snow conditions (Essery et al., 2013; Lafaysse et al., 2017).

### 2.5.2 Multivariate DA experiments

Several experiments have been carried out with the aim of assessing the performances of the multivariate SIR-PF scheme under different configurations, as listed in Table 5.

In the first experiment [M_Exp] the efficiency of the DA scheme in updating the model snowpack states is tested by assuming the meteorological data as the only source of uncertainty. The snow observations are jointly assimilated every 3 hours to ensure an efficient exploitation of the high frequency in-situ measurements supplied by the automatic stations at the

analysed snow experimental sites.

The second experiment [MP_Exp (1)] aims at assessing the impact of the perturbation of the model parameters on the filter performance. Indeed, the introduction of the parameters uncertainty is supposed to contribute to limiting the sample impoverishment by enlarging the ensembles spread, whose size strongly impacts the filter performance. Therefore, along with the perturbation of meteorological inputs, each particle independently evolves according to its own specific set of

parameters ruling the model physical dynamics, whose equations remain unchanged, however. Thanks to this approach, the degeneracy of the model ensembles is limited via resampling and the sample impoverishment is prevented through the parameters perturbation. However, as stated by Salamon and Feyen (2009), when dealing with parameters uncertainty it is important to consider that the model response to a change in parameters does not have an immediate effect on the simulated state. This issue can be overcome by giving the model a sufficiently large response time between following system updates.

The assimilation frequency is therefore reduced to every 24 hours. This choice is intended to guarantee a higher model response time without omitting a large number of observed snow data.

The sensitivity of this last configuration of the DA scheme to the measurement frequency is investigated by assessing how the filter performance is affected when limiting the observational dataset of the CDP site from daily to bi-weekly SWE measurements [MP_Exp (2)].

With the aim of reducing the system sensitivity to the availability of snow mass observations, the fourth experiment [MPP_Exp] tests the potential of using indirect information on SWE and snow density, whose measurements are generally time-consuming and often not available for real-time applications. According to the methodology proposed by Jonas et al. (2009), an additional empirical snow density model is introduced to reliably determine indirect sampling of SWE state from snow depth measurements through a parameterization of snow density, depending on four main factors: seasonality, observed snow depth, site altitude and location. With the aim of evaluating the reliability of the resulting estimates, a qualitative comparison analysis is performed with respect to the observations available at the Swiss and Italian measurement sites, as shown in Figure 4. Except for some sporadic winter seasons, generally the estimates of SWE and snow density well fit the observed snowpack dynamics, as demonstrated by a good agreement with the ground-based measurements. However, it is noteworthy that the estimate of snow density features is more challenging for shallow snow depths, since a high variability can range from low-density new snow in early winter to high-density slush during springtime (Jonas et al., 2009).

When dealing with a multivariate assimilation of several observed variables, it is of key importance to investigate whether the ensemble size can be sufficient to efficiently describe the high-dimensional assimilation scheme. With the aim of addressing this critical issue, after identifying the most proper DA configuration for each experimental site, according to the local features and the availability of observed data, the system sensitivity to the ensemble size is investigated by testing 100-, 200- and 500-particles ensemble simulations [nP_Exp]. This last experiment is performed by considering a sample of one winter season for each experimental site: snow seasons 2007/08 at CDP, 2001/02 at WFJ, and 2012/13 at TGN site. The impact of the variation in the particles number on the filter performance is analysed by evaluating both the ensembles spread and the ensemble effective size at the resampling step, namely the number of selected particles having significant likelihood values with respect to the total ensemble size.

It is worth considering that the assimilation time is properly defined to not neglect any snow-mass related observation through both 3- and 24-hours assimilation frequencies. Indeed, these quantities are measured at a fixed scheduled time, namely at 12 am at the French and Italian sites and at 11 am at the Swiss station.

### 2.5.3 Probabilistic open loop control run

With the aim of properly analysing the skill of the multivariate DA scheme, each experiment is evaluated through the comparison with control open loop (without DA) ensemble simulations forced by perturbed meteorological data (Ens_OL) (Sect. 2.4.1).

After verifying that the introduction of the stochastic noise does not affect the observed inputs on average at any site, firstly the aim is to assess the impact of the meteorological perturbation on the ensemble snowpack simulations, without considering the assimilation of snow data. Indeed, since the strong system nonlinearities make the model response to the

inputs perturbation hardly predictable, it is important to verify that no unexpected biases occur with respect to the deterministic control run. Secondly, it is important to consider how the perturbation of the meteorological data succeeds in realistically depicting the uncertainty of snow model simulations.

### 2.5.4 Evaluation metrics

The results of this study are shown and discussed in terms of SWE, snow depth and surface temperature. The assessment of both SWE and snow depth simulations allows to indirectly evaluate also the model dynamics of snow density (Jonas et al., 2009). Furthermore, the impact of the filter updating on the system energy balance is analysed through the evaluation of the simulations of the surface temperature.

In order to properly assess the filter performance, each experiment is evaluated through a deterministic statistical index
related to the ensemble mean simulations, and an ensemble-based probabilistic skill metrics. These multi-year evaluation metrics are computed by considering the whole datasets of measurements excluding the snowless periods, which conservatively start after the latest melt-out date recorded at each experimental site.

The Kling-Gupta Efficiency (KGE) coefficient (Gupta et al., 2009) allows to analyse how the assimilation of snow observations succeeds in properly updating the model simulations, on average:

$$KGE = 1 - \sqrt{(r-1)^2 + (a-1)^2 + (b-1)^2} \qquad (11)$$

where:

- r is the linear correlation coefficient between the mean ensemble simulations and observed values;
- a is the ratio of the standard deviation of mean ensemble simulations to the standard deviation of the observed ones, i.e. an estimate of the relative variability between simulated and observed quantities;
- b is the ratio of the mean of mean ensemble simulations to the mean of observed ones, i.e. a measure of the overall bias.

The optimal KGE value is ideally equal to 1, revealing that the ensemble mean simulations succeed in well catching the observed values (theoretically, r=1, a=1, b=1).

The Continuous Ranked Probability Skill Score (CRPSS) is evaluated to assess changes to the overall accuracy of the
ensemble simulations of each experiment (CRPS) by considering the open loop ensemble (Ens_OL) control run as the reference one (CRPS$_{ref}$), according to the formula:

$$CRPSS = 1 - \frac{CRPS}{CRPS_{ref}} \qquad (12)$$

The Continuous Ranked Probability Score (CRPS) measures the error in the cumulative probability distribution computed from the ensemble members relative to observations (Hersbach 2000):

$$CRPS = \int_{+\infty}^{+\infty} \left( P_{ens}(x) - P_{obs}(x) \right)^2 dx \qquad (13)$$

The smaller the CRPS value, the better the probabilistic simulation is (perfect score equal to 0). Conversely, the optimal value of CRPSS is equal to 1 and negative values indicate poorer performances with respect to the reference control run.

## 3. RESULTS AND DISCUSSION

### 3.1 Open loop ensemble simulations

To investigate the impact of the meteorological stochastic perturbations, 100-ensemble snowpack simulations forced by as many different meteorological conditions are analysed. For the sake of concision and clarity, a representative winter season

is shown for each site (Figure 5). The ensembles spread reveals possible over- and underestimation of the ensemble model simulations as direct consequence of the perturbation of the forcing data. Since the meteorological perturbations are unbiased, this issue is mainly due to the nonlinearity of the involved physical processes. Therefore, it is noteworthy to observe that, even though the time series of the deterministic control open loop run are generally included within the ensemble envelop, they differ from the ensemble mean simulations. The variance of the mass-related ensembles is generally

the largest at the end of the winter season, when the perturbation of energy-related forcing variables (i.e. air temperature, shortwave radiation) leads to well-spread melt-out scenarios resulting from the difference in melt timing (i.e. some particles have just started to melt and some others have already disappeared). During the winter season, the spread of SWE ensembles is increased whenever a snowfall event occurs due to the uncertainties in the precipitation rates allowing to provide the mass balance of each model realization with different input of snowfall rate. Of course, sites climatology (e.g. frequency of

snowfall events) strongly impacts the resulting ensemble variance. A significant variance of the surface temperature ensembles is ensured by its high sensitivity to the inputs uncertainty, namely the perturbation of the air temperature and shortwave radiation, which directly impacts the snowpack energy balance. However, it is important to consider that some threshold processes involved within the snow dynamics model (e.g. disappearance of the surface snow layer, limitation of state variable within physical ranges) can be counter-productive in enlarging the ensembles spread. Figure 6 shows the

Talagrand diagrams (Hamill, 2001) of the Ens_OL SWE simulations at CDP site. The deterministic open loop predictions are properly included within the ensemble envelop. However, the shape of the Talagrand distribution reveals an underestimated trend of the Ens_OL simulations, on average, with respect to the deterministic ones. This issue is more prominent when considering the SWE observations, as proven by the peak in the rank histogram. The distribution of the SWE simulations shows that the ensemble is under-dispersive, which warns about the limited representativeness of the

ensemble spread achieved by perturbing the meteorological inputs.

### 3.2 Multivariate DA simulations with perturbed meteorological input data

With respect to the control run (Ens_OL), the multi-year KGE values of the multivariate DA simulations relying on the perturbation of the meteorological data [M_Exp] reveal the filter effectiveness in updating snow depth simulations (Figure 7).

Conversely, the update of SWE model predictions is more challenging. At the French station, the assimilation of snow data actually leads to a worsening of the quality of the SWE simulations with respect to the probabilistic control run. A slight improvement is observed at the Swiss and Italian sites, where the filter updating benefits from a larger spread of the SWE

simulations ensured by a higher frequency of snowfall events, on average. Even though the filter well succeeds in enhancing the simulations of surface temperature at CDP site, the snowpack thermal state at TGN and WFJ is poorly affected by the assimilation of snow data.

To better understand and properly assess the results, it is important to stress some key conditions exerting the most influence on the filter effectiveness. One of the main ruling issues is the scale of the model ensemble spread. A well-spread ensemble makes the filter efficient in weighting the particles, since they are properly discriminated through different likelihood values (Ades and Van Leeuwen, 2013). Alongside this issue, it is of critical importance how the particles are placed on average with respect to the measure of the corresponding variable. The most conducive condition calls for well-spread ensembles enclosing their corresponding observations. However, the spread of the model ensembles turns out to be the overriding condition. Indeed, even if the model predictions are biased, a large ensemble spread can allow to progressively stretch the simulations towards the observed system state through subsequent proper updates.

When dealing with a multivariate DA scheme, the fulfilling of these conditions is even more challenging. In such an application, the filter is designed to select the particles best describing the observed system state with respect to all the available observations at the assimilation time step. Therefore, with respect to a univariate DA scheme, here the filtering procedure is more heavily constrained, depending on how many observations are provided.

Even though the effects of the ensemble degeneracy can be reduced through the resampling procedure, the perturbation of the meteorological data turns out to be not sufficient to prevent the sample impoverishment within two following assimilation time steps (e.g. SWE ensemble when no snowfall event occurs). It is noteworthy that a decrease in the ensemble spread, even just of one variable, can affect the overall resampling procedure. As previously explained, this limitation is even intensified by the physics of the snowpack model, whose threshold processes can weaken the effect of inputs perturbation.

Another further issue is the difference in the measurements frequency of the variables to assimilate. At each assimilation time step the Gaussian likelihood function is n-dimensional depending on the number of the observed variables. Thus, the particles weighting is carried out considering their likelihood in relation to the available measurements at that time. This dynamic entails that the resampling procedure can be more strongly conditioned by the observations having a higher measurement frequency (e.g. hourly or sub-hourly measurements of snow depth). Thus, possible misleading updates of the variables less frequently observed can occur, since they are updated without taking into account particles likelihood with respect to their own lacking observations (e.g. daily or bi-weekly measurements of SWE). For instance, when a measure of snow depth is provided, no observational information on SWE can be properly retrieved except its indirect estimate if snow density data are available. Otherwise, the filter can fail in consistently updating the overall snow mass-related state, since a lot of possible combinations of SWE and snow density can well fit the observed snowpack depth. In terms of filter efficiency, this means that when only a snow depth observation is provided the filter looks for particles having the higher likelihood with respect to this snow quantity, regardless the SWE and snow density states. Nevertheless, it is not unlikely that several particles have the same likelihood because the combination of even strongly-biased values of SWE and snow

density can well fit the observed snow depth through an offsetting effect among these variables. This is the main reason explaining the higher filter performance in terms of snow depth with respect to the SWE simulations.

The multivariate DA simulations allow to point out two main limitations of this application. Firstly, the ensemble spread needs to be enlarged in order to improve the filter efficiency in well weighting and resampling the ensemble particles. Moreover, the effect of the difference in measurements frequency of the assimilated variables has to be limited in order to consistently update the snow mass balance.

## 3.3 Multivariate DA simulations with perturbed model parameters

Figure 7 shows the multi-year KGE values of the multivariate DA simulations resulting from the implementation of the model parameters perturbation [MP_Exp]. With respect to the previous experiment considering the meteorological data as the only source of uncertainty (Sect. 3.2), the introduction of the parameters perturbation allows to heavily improve the filter efficiency at updating the model SWE simulations at CDP site. Table 6 shows this significant enhancement in terms of CRPSS, whose negative value for the M_Exp SWE simulations at the French station increases up to 0.74. It is noteworthy that the parameters perturbation does not only ensure a sizeable enlargement of the ensembles spread but it also allows to better estimate the model parameters on average. Indeed, while the resampling of the state variables allows to consistently update the system state at the assimilation time step, the modelling of snowpack dynamics between two following assimilation time steps benefits from the parameters resampling, which enables to take better account of the parameters seasonality (e.g. melting period). Figure 8 provides a case in point, showing how the parameters perturbation impact on the filter effectiveness in terms of both the ensemble spread and its positioning with respect to the observation, at the same assimilation time step.

At the French site, the daily SWE measurement frequency ensures an effective resampling of the mass-related parameters, namely snow roughness and snow viscosity. Furthermore, the retention of satisfying performance of the filter in terms of snow depth on average suggests a beneficial impact on the snow density dynamics. Conversely, the SWE simulations at the Swiss station do not benefit from the introduction of the parameters perturbation (Table 6). This limitation is mainly due to the lower biweekly frequency of the SWE measurements, with respect to the French case study. At WFJ site, at the daily assimilation time steps when no SWE observation is available, the estimate of the particles likelihood cannot rely on observational information on the snow mass-related parameters (e.g. SWE, snow density). Therefore, it is not unlikely that the resampling procedure leads to suboptimal values of the mass-related parameters. Moreover, the enlargement of the ensembles spread ensured by the parameters perturbation entails a higher probability of selecting particles having SWE values even farther from the actual state with respect to the simulations of the deterministic control run, when no direct SWE observed data are provided. This thesis is supported by the annual results obtained at the TGN site (not shown here), where the multivariate DA scheme allows to consistently update the SWE simulations when the daily average SWE measurements are available, namely throughout the winters 2013-14 and 2015-16. During the other two snow seasons, when biweekly SWE observations are assimilated, the filter does not succeed in improving model predictions.

The filter updating is not as effective for the simulation of surface temperature, especially at the Swiss and Italian sites (Table 6). This suboptimal performance is mainly addicted to the concurrence of several factors. Firstly, the quicker dynamics of the daily thermal cycle make the temperature simulations more sensitive to the reduction in assimilation frequency, with respect to the other variables. Secondly, even though the filter succeeds in daily updating the system thermal

state, the parameters values resulting from the resampling procedure do not ensure a long-lasting effect on the temperature trend between two following assimilation time steps. Indeed, since the parameters are resampled according to their representativeness at the assimilation time step, their values are not likely to be the optimal ones to well catch the succession of diurnal and nocturnal peaks.

Although the parameters perturbation ensures an enlargement of the ensembles spread, which is one of the constraining

conditions to ensure the filter effectiveness, the quality of the multivariate DA simulations strongly depends on the reliability of the parameters resampling, which requires direct observational information to properly estimate the more likely parameters values.

### 3.4 Sensitivity analysis of the multivariate DA scheme to the SWE measurement frequency

With the aim of investigating the system sensitivity to the SWE measurement frequency, an experiment is performed at the

CDP station to assess how the reduction from daily to biweekly SWE observed data affects the 24-hours multivariate DA simulations [MP_Exp (2)]. Obviously, a reduction in measurement frequency is expected to reduce the impact of the filter updating on the model simulations. However, when dealing with a multivariate DA scheme, the imbalance among the measurement frequency of the assimilated variables can lead to a further side-effect hindering the parameters estimate due to the largest impact of the more frequently observed variables (e.g. snow depth, surface temperature) on the particles

weighting. Figure 9 shows the ensembles of snow viscosity and snow roughness resulting from the assimilation of daily and biweekly SWE observations throughout the winter season 2001-2002. A divergence between the two ensemble time series is clearly detectable on average, especially in terms of snow viscosity. The suboptimal estimate of the mass-related parameters based on biweekly SWE measurements leads to a worsening of model predictions with respect to the control run, as shown in Figure 10. Conversely, the filter effectiveness is not affected in terms of snow depth thanks to offsetting effects between

SWE and snow density simulations.

### 3.5 Multivariate DA simulations with proxy information of snow mass-related variables

Even though the introduction of the parameters uncertainty well succeeds in enlarging the ensembles spread, the resampling procedure of both states and parameters turns out to be even counter-productive when it is not properly conditioned by observed data of ruling snow mass-related variables. The implementation of the additional snow density model providing

proxy information on the mass-related snow variables at the Swiss and Italian sites allows to optimize the parameters resampling, as revealed by the outperforming statistical scores of the SWE simulations, especially at WFJ station (Figure 7, Table 6). Conversely, no prominent effects are observed in terms of surface temperature and snow depth.

The reduction in assimilation frequency necessarily leads to omitting large quantities of observed data. With the aim of preventing this limitation, the approach proposed by Salamon and Feyen (2009) has been tested. According to this method, each particle is assigned the median of the weights evaluated at all observation time steps within the 24-hrs response time interval. Although this approach allows to make full use of the available measurements, a more intensive use of proxy information on the snow mass-related variables makes the filter effectiveness more affected by the quality of the estimates, with resulting heterogeneous filter performance over the analysed datasets.

## 3.6 Sensitivity analysis of the multivariate DA scheme to the ensemble size

Figure 11 shows the main results of the experiment aiming at assessing the system sensitivity to the ensemble size [nP_Exp]. When evaluating the effective sample size after each filter update, expressed as percentage of selected particles with respect to the total number of the ensemble size (i.e. 100, 200, and 500 particles), it is noteworthy that an increase in the particles number generally does not result in a significant increment of the percent effective sample size. On average, this quantity ranges around 50% for the CDP and WFJ station, and up to 60% at the Italian experimental site. Furthermore, when assessing the impact of the variation in the particles number on the filter performance in updating the model simulations (Table 7), the resulting multi-year KGE values do not reveal a systematic improvement in the simulations reliability as the ensemble size increases.

A low system sensitivity to the ensemble size is also clearly proven by the slight impact of the change in the particles number on the ensembles spread (Figure 11). Indeed, the increase of the ensemble size generally does not ensure a proportional enlargement of the particles spread, expect for the snow depth, whose 500-particles ensemble simulations reveal a slightly larger spread at CDP and TGN site.

Despite being a multivariate application of the PF-based scheme, the results of this experiment mainly demonstrate that a 100-particles ensemble can be assumed as sufficiently representative for a point-scale application.

## 4. CONCLUSIONS

This study investigated the potentials of a SIR-PF scheme for a multivariate assimilation of snow data at three experimental sites in the Alps. Even though PF technique proved its capability of properly handling the strong system nonlinearities of snow modelling, several challenging issues need to be addressed.

When dealing with a multivariate DA application, the sample impoverishment is more likely to occur with respect to the univariate case, since the filter is designed to strictly select the particles having the highest likelihood with respect to all the observed state variables. The perturbation of the meteorological forcing data has turned out not to be sufficient to restore the ensembles spread within two following 3-hrs assimilation time steps, with resulting poor filter performance, especially in terms of SWE. In order to prevent this undesired condition, further stochastic noise has been introduced through the parameters perturbation, with a reduction of the assimilation frequency to every 24 hours to ensure a sufficient model

response time. At CDP site, where SWE observations are available with a comparably high (daily) frequency with respect to the other assimilated variables, this filter setup outperforms the PF-based simulations considering the meteorological data as the only source of uncertainty. Conversely, at the Swiss and Italian stations, the benefit of introducing additional stochastic noise through the parameters perturbation is overcome due to the lower (biweekly) measurement frequency of the snow

mass-related quantities. Indeed, even though the parameters perturbation succeeds in enlarging the model ensembles, the system has revealed a prominent sensitivity to the difference in measurement frequency of the assimilated variables, which hinders the filter effectiveness in consistently updating the modelled snow quantities. Actually, the more frequently measured snow variables (e.g. snow depths) strongly condition the mass-related parameters resampling (i.e. snow viscosity and snow roughness), which can result in possible suboptimal values.

Where snow mass-related observations are less frequently available, the assimilation of indirect estimates has revealed a remarkable potential to make up for the lack of information within the resampling procedure. At WFJ and TGN sites, the introduction of an additional model providing proxy data of snow density and SWE has allowed to improve the consistency of the filter updating.

The filter has turned out to be less effective in updating the simulation of surface temperature, mainly due to the quick

dynamics of the daily thermal cycle which entails a short-lasting representativeness of the parameters values to properly describe the diurnal and nocturnal peaks.

In this point-scale application of the multivariate SIR-PF scheme, the system has revealed a low sensitivity to the ensemble size, thereby proving that 100 particles can be suited to represent the high dimensionality of the system. However, when

modelling at larger scale, the sensitivity to the ensemble size needs to be deeply investigated, especially for multilayer snowpack models.

Another critical issue of key importance for spatialized applications is the combined perturbation of meteorological data and model parameters, since the high dimensionality of the modelling systems necessarily requires a consistently large stochastic noise to ensure the effectiveness of the filter updating. Especially during periods when no snowfall event occurs, the

meteorological uncertainty is assumed not to be sufficient to ensure a well-representative range of possible snowpack states. Furthermore, in multivariate PF-based applications for snow modelling, it is noteworthy to investigate the potential of introducing indirectly estimated information on those state variables not directly measured. Indeed, the lack of measures in ungauged or poorly gauged areas can hinder the application of a PF-based scheme, whose resampling procedures highly benefit from comprehensive information on the snowpack state to properly detect those particles having a higher overall

likelihood value. This issue is even more compelling when dealing with a detailed physically-based snowpack model.

Nevertheless, several issues require further detailed analysis. The physical consistency of the meteorological ensembles needs to be improved. Indeed, the methodology does not take into account the correlations among the perturbed forcing variables and, moreover, the specific error statistics characterizing the perturbations are to be specifically evaluated at each analysed site. Even though the evaluation of the likelihood function for high-dimensional systems becomes more challenging

(Margulis et al., 2015), the potential of using empirical likelihood variants should be extensively investigated to assess the impact of using a Gaussian distribution (Leisenring and Moradkhani, 2011; Thirel et al., 2013). Furthermore, an interest is focused on testing other resampling techniques with the aim of analysing how the resampling procedure affects the filter effectiveness (Moradkhani et al., 2005; Weerts and El Serafy, 2006; Salamon and Feyen, 2009).

*Data availability*

The snow and meteorological data from Col de Porte site are made freely available by Météo-France both on the PANGAEA depository (doi:10.1594/PANGAEA.774249) and on the public ftp server ftp://ftp-cnrm.meteo.fr/pub-cencdp/. The Weissfluhjoch dataset provided by the WSL Institute for Snow and Avalanche Research SLF can be obtained via IDAWEB
(https://gate.meteoswiss.ch/idaweb) as well as from the Environmental Data Portal ENVIDAT (envidat.ch/dataset/10-16904-1). The dataset of Torgnon site is available on the European Fluxes Database (www.europe-fluxdata.eu/).

*Author contributions.*

This work is part of the G. Piazzi's PhD thesis, supervised by S. Gabellani, L. Campo. G. Thirel supervised the research
activities during the visiting period of G. Piazzi at the Catchment Hydrology Research Group of the HYCAR Research Unit (IRSTEA). All the authors have collaborated to the technical development of the modelling system and the manuscript writing.

*Acknowledgements.*

The research has been conducted in the framework of the collaboration between the HYCAR Research Unit of the National Research Institute of Science and Technology for Environment and agriculture (IRSTEA) and CIMA Research Foundation. We wish to thank both institutions for supporting and promoting these research activities. The work has been also supported by Regione Autonoma Valle d'Aosta and the Next-Data Project. The authors would like to acknowledge Météo-France, the WSL Insitute, and the Environmental Protection Agency of Aosta Valley for providing the datasets used in this study. We
are grateful to all the staff working at Col de Porte, Weissfluhjoch and Torgnon experimental sites, who ensure high-quality snow observations for many years. We would like to address special thanks to all the colleagues for the fruitful discussions on this matter. We are also grateful to Matthieu Lafaysse and the anonymous reviewer for helping to significantly improve this manuscript through their constructive remarks.

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

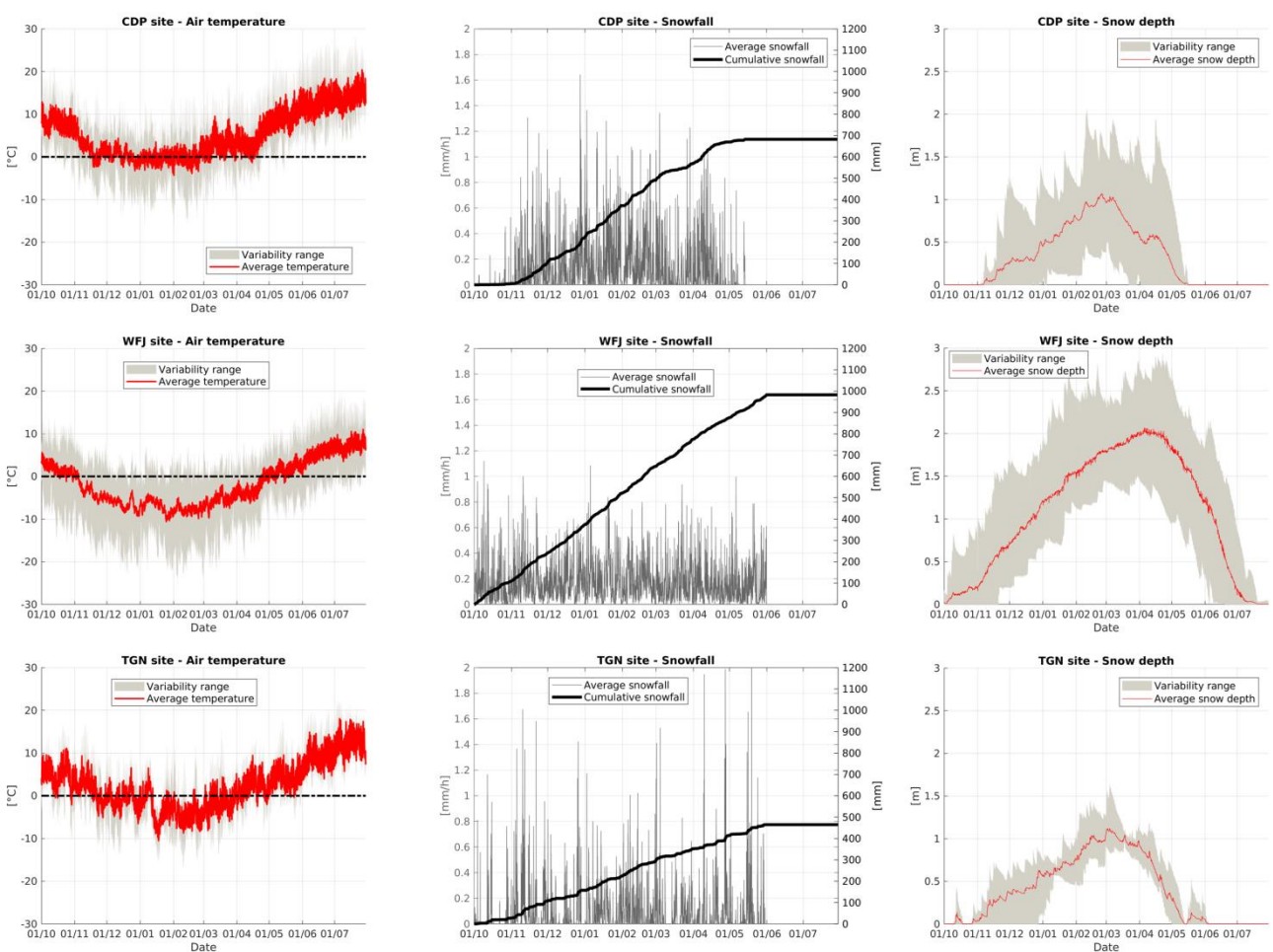

**Figure 1: Meteorological characterization of CDP site (first row), WFJ site (second row), TGN site (third row) - Air temperature (left column), snowfall rates (middle column; at WFJ and TGN sites snowfall rates have been estimated according to Froidurot et al., 2014) and snow depth (right column) throughout an average snow season (early October – early July) throughout the overall datasets.**

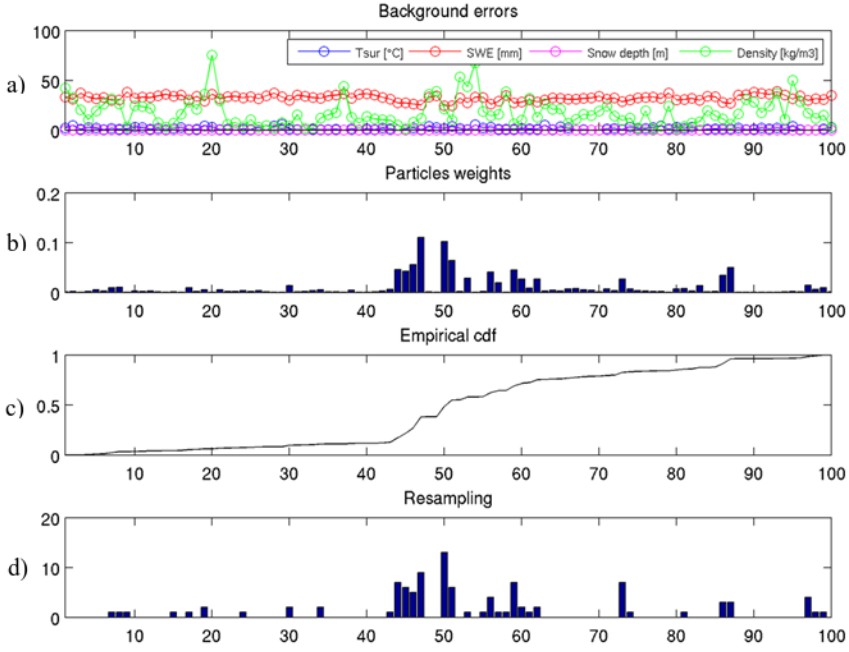

**Figure 2: SIR-PF scheme for multivariate DA: (a) Open circles are the background errors for surface temperature (in blue), SWE (in red), snow depth (in magenta) and snow density (in green). (b) Importance weights as a function of the particles indices. (c) Empirical cdf of the weights. (d) Number of resampled particles as a function of the particles indices.**

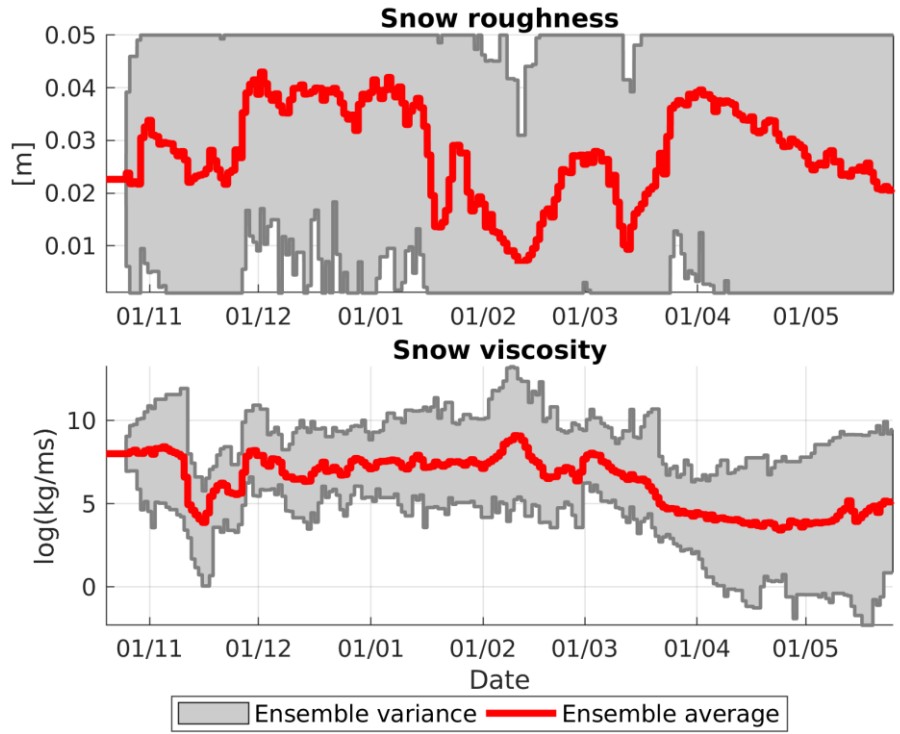

**Figure 3: Uncertainty of model parameters – CDP site – Winter season 2010-2011. The panel shows the ensembles seasonality of snow roughness and snow viscosity.**

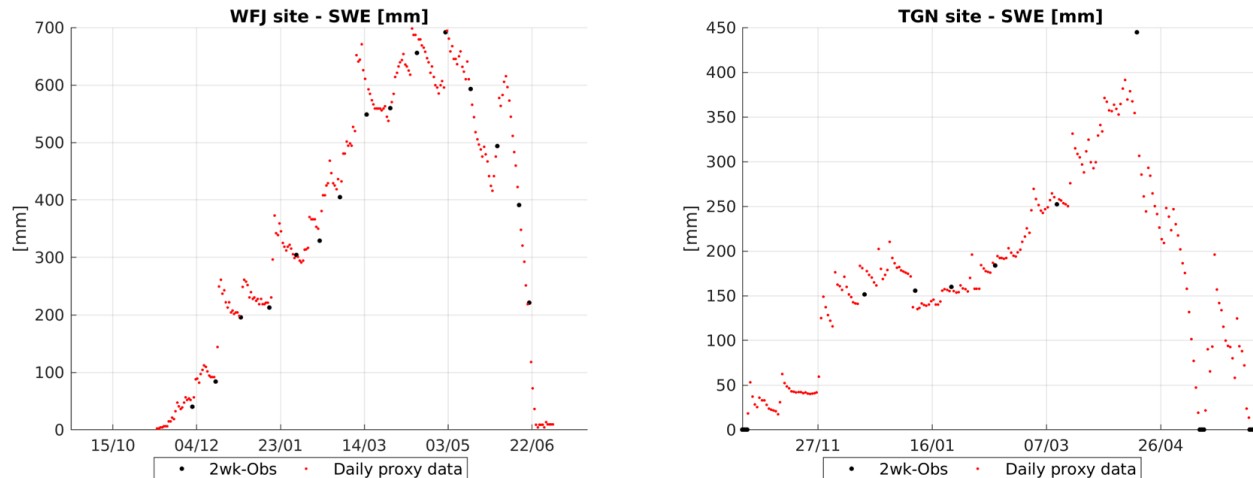

**Figure 4: Additional snow density model –Comparison between SWE measurements and indirect proxy estimates - WFJ site, winter season 2005/06 (on the left); TGN site, winter season 2012/13 (on the right).**

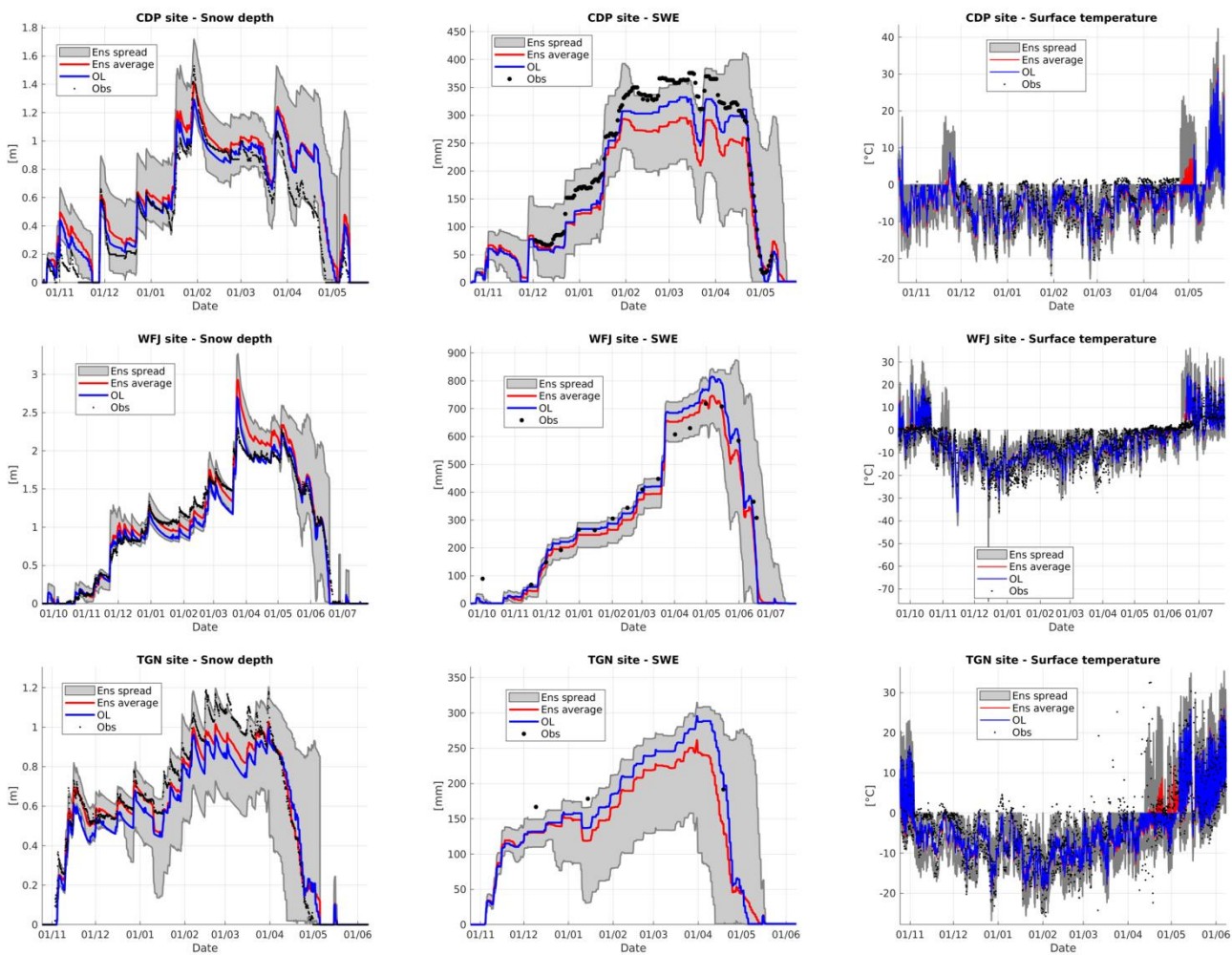

**Figure 5: Impact of the meteorological uncertainty - Ensemble simulations of snow depth (left column), SWE (middle column), and surface temperature (right column) – CDP, winter season 2003-2004 (first row); WFJ, winter season 2001-2002 (second row); TGN, winter season 2014-2015 (third row).**

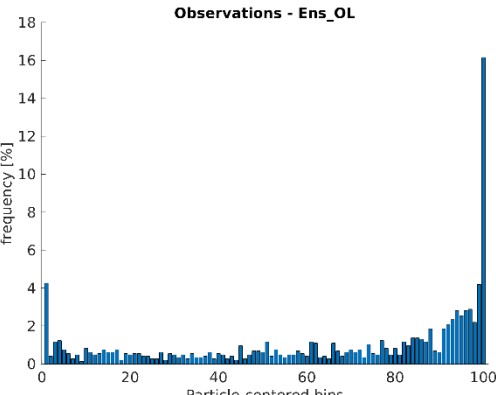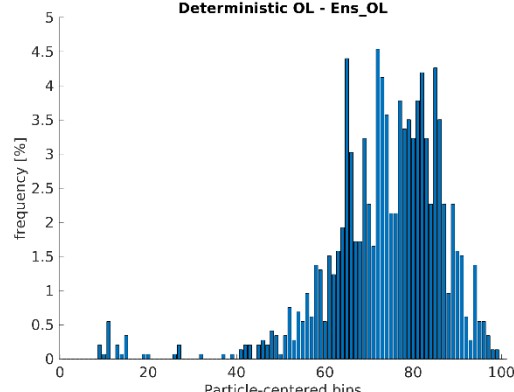

**Figure 6: Talagrand diagram – Analysis rank histogram of SWE ensemble open loop (Ens_OL) simulations considering SWE observations (on the left) and the SWE deterministic open loop predictions. CDP dataset (10 winter seasons).**

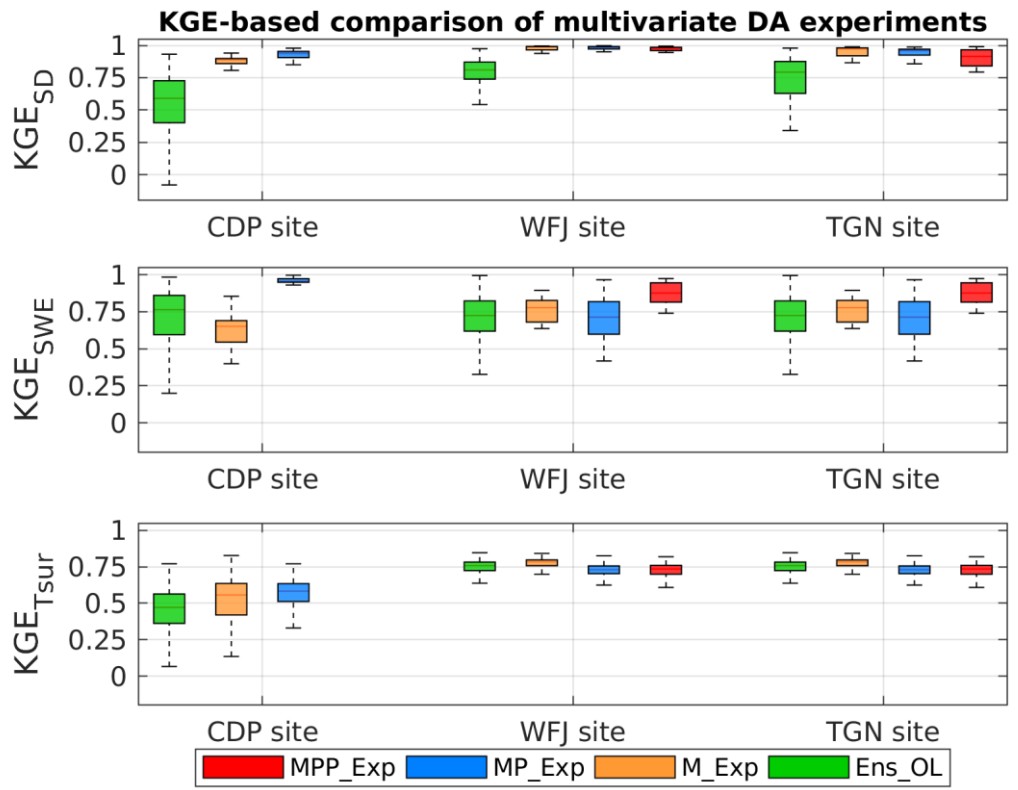

**Figure 7: Multivariate DA experiments – Multi-year KGE values of snow depth (SD), SWE, and surface temperature (T$_{sur}$) simulations. The Ens OL, M_Exp, MP_Exp (1), and MPP_Exp statistical scores are in green, orange, blue and magenta, respectively. The bottom and top edges of each box indicate the 25$^{th}$ and 75$^{th}$ percentiles, respectively. The line in the middle of each box is the median. The whiskers extending above and below each box indicate the 5$^{th}$ and 95$^{th}$ percentiles.**

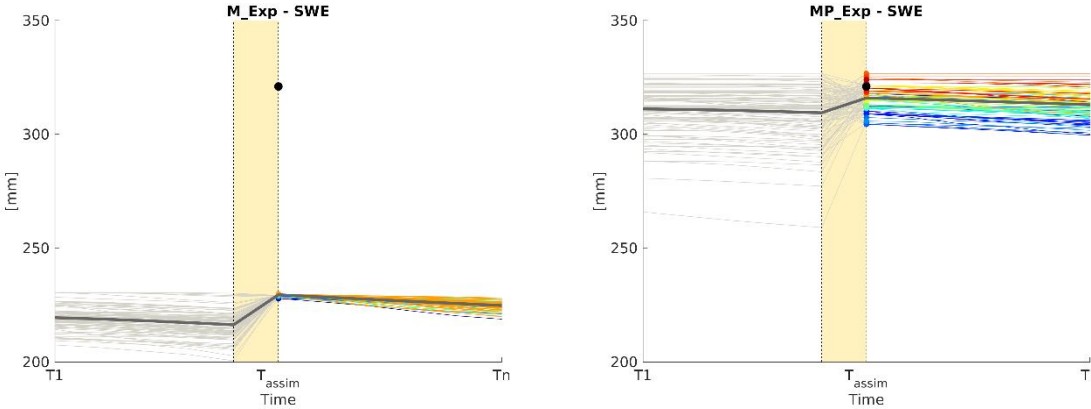

**Figure 8: Particles resampling at an assimilation time step of SWE observation – M_Exp vs MP_Exp – In grey are the ensemble members, whose ensemble mean is represented thicker; the black dot is the observation; the resampled particles are multicoloured.**

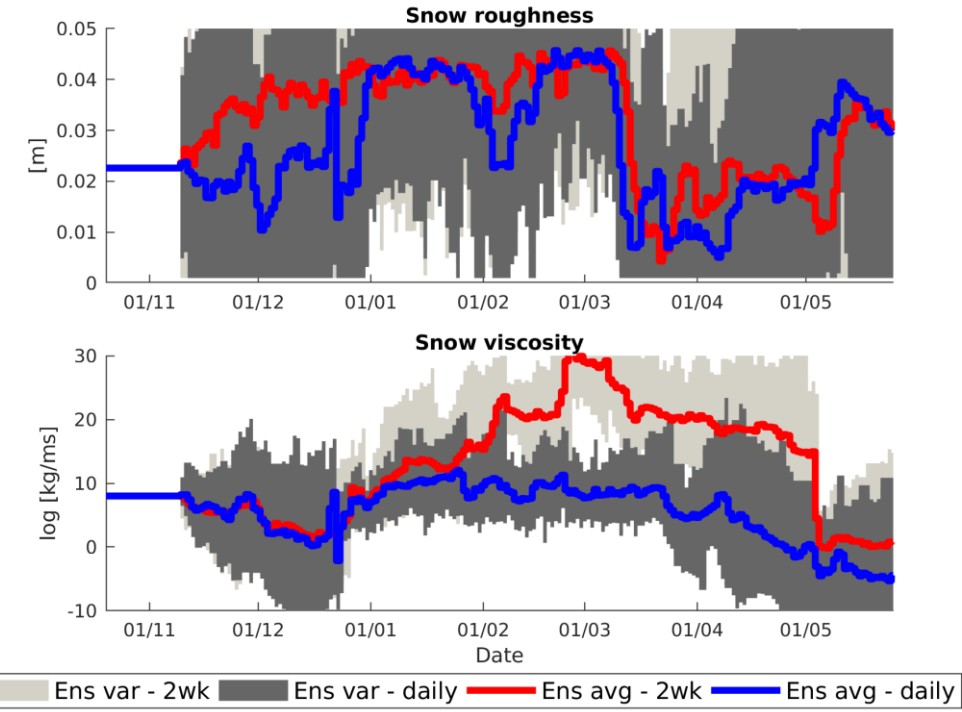

**Figure 9: Sensitivity analysis of the multivariate DA scheme to SWE measurement frequency at CDP site, winter season 2001/02 – Parameters ensembles: snow roughness (on top) and snow viscosity (second row) resulting from the assimilation of daily (average trend in blue) and biweekly (red) SWE observations.**

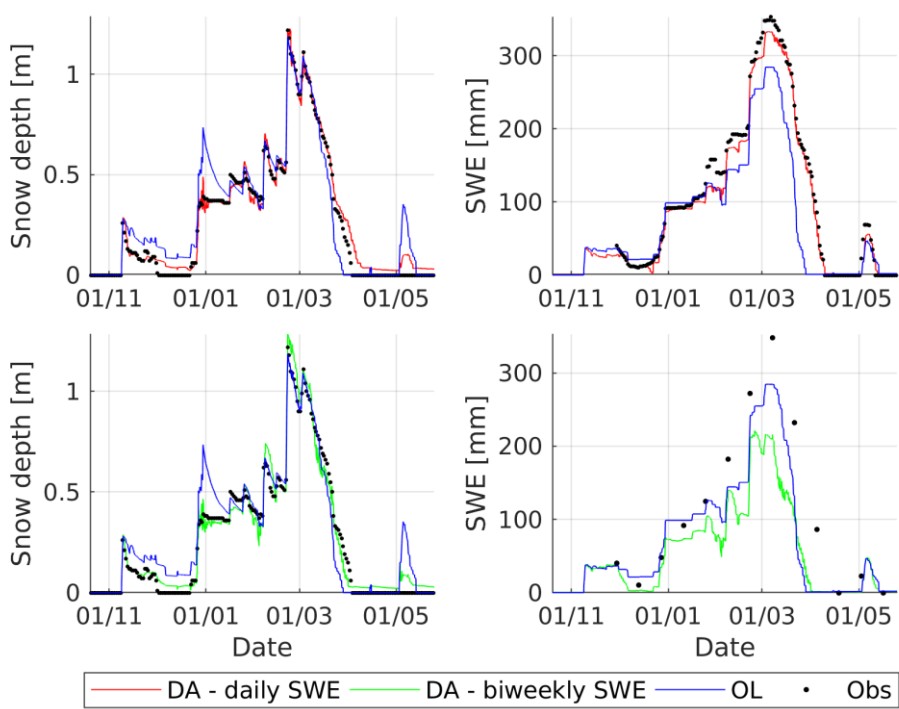

**Figure 10: Sensitivity analysis of the multivariate DA scheme to SWE measurement frequency at CDP site, winter season 2001/02 – Mean ensemble simulations of snow depth (left column) and SWE (right column) for daily (first row) and biweekly (second row) SWE measurements.**

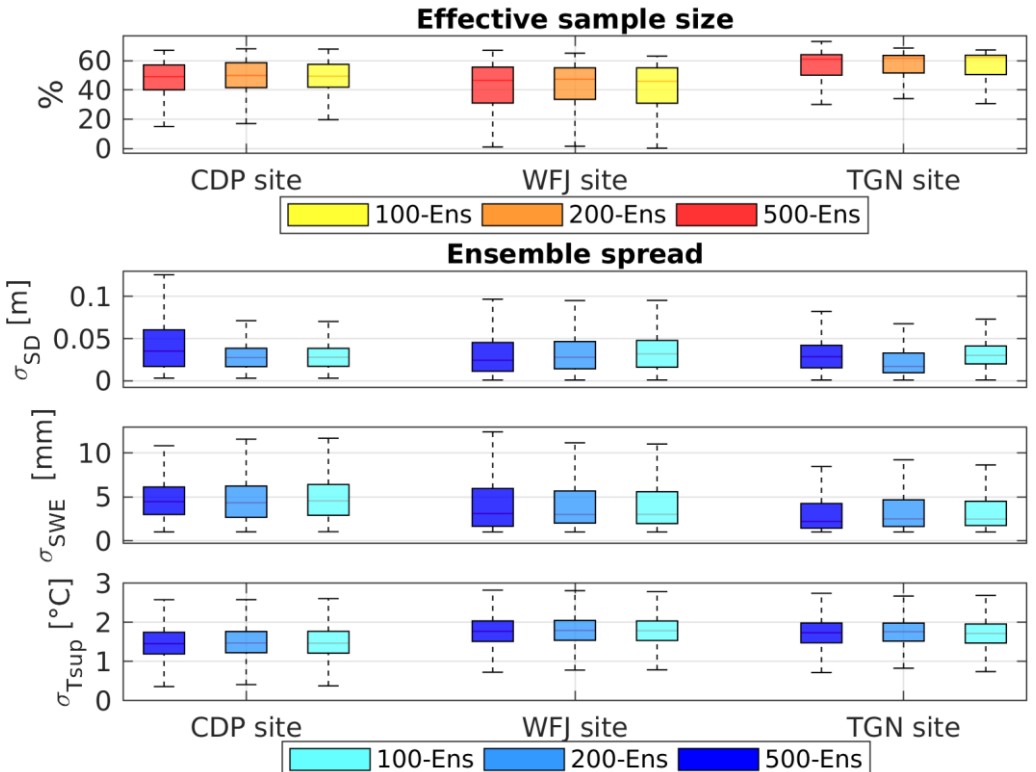

**Figure 11: nP_Exp – Effective sample size and ensemble spread as a function of the ensemble size.**

| Variable | | Unit | Distribution | μ | σ | τ [h] | Lower limit | Upper limit |
|---|---|---|---|---|---|---|---|---|
| Air temperature | $T_a$ | [°C] | Normal | 0 | 0.9 | 4.8 | - | - |
| Relative humidity | RH | [%] | Normal | 0 | 8.9 | 8.4 | 0 | 100 |
| Solar radiation | SW | [W/m²] | Normal | 0 | min(SW, 109.1) | 3.0 | 0 | - |
| Precipitation | P | [mm] | Lognormal | -0.19 | 0.61 | 2.0 | 0 | - |
| Wind speed | V | [m/s] | Lognormal | -0.14 | 0.53 | 8.2 | 0.5 | 25 |

**Table 1: Error statistics for the generation of the meteorological ensembles (Magnusson et al., 2017).**

| | Parameter | | Nominal value | Range |
|---|---|---|---|---|
| 1. | Snow roughness | [mm] | 0.0226 | $[0.001 - 0.05]$ |
| 2. | Snow viscosity | [kg/ms] | $10^8$ | $[7 \cdot 10^7 - 10^{12}]$ |
| 3. | Albedo parameter $\tau_\alpha$ | [s] | $10^7$ | $[0.52 - 1.55 \cdot 10^7]$ |
| 4. | Albedo parameter $\tau_m$ | [s] | $3.6 \cdot 10^5$ | $[1.73 - 5.2 \cdot 10^5]$ |
| 5. | Albedo parameter $S_0$ | [mm] | 10 | $[2 - 15]$ |

**Table 2: Sensitivity analysis - Selected model parameters.**

| Site | Dataset size | Snow seasons | | Reference |
|------|--------------|------|----|-----------|
|      |              | from | to |           |
| CDP | 10-years | 2001/2002 | 2010/2011 | Morin et al., 2012 |
| WFJ | 16-years | 1999/2000 | 2014/2015 | WSL Institute for Snow and Avalanche Research SLF, 2015b |
| TGN | 4-years | 2012/2013 | 2015/2016 | Galvagno et al., 2013 |

**Table 3:** *Snow datasets.*

| | | | Assimilated variables | | |
|---|---|---|---|---|---|
| Site | $T_{snow}$ | SWE | α | $SD$ | $\rho_{snow}$ |
| | [°C] | [mm] | [-] | [m] | [kg/m³] |
| | $\sigma_{obs} = 2°C$ | $\sigma_{obs} = 10$ mm | $\sigma_{obs} = 0.15$ | $\sigma_{obs} = 0.05$ m | $\sigma_{obs} = 50$ kg/m³ |
| CDP | hourly | daily | hourly | hourly | **daily** |
| WFJ | 30-min | bi-weekly | daily | 30-min | **bi-weekly** |
| TGN | 30-min | daily (2013/14, 2015/16); **bi-weekly (2012/13, 2014/15)** | daily (2012/13) | 30-min | **daily (2013/14, 2015/16)**; bi-weekly (2012/13, 2014/15) |

**Table 4: Measurement frequency of the assimilated variables at each experimental site: snow surface temperature ($T_{snow}$), SWE, albedo (α), snow depth (SD) and snow density ($\rho_{snow}$) – Observational uncertainties ($\sigma_{obs}$) are reported - Variables indirectly estimated are highlighted in bold.**

| Experiment | Ensemble generation | Assimilation period | Specific details |
|---|---|---|---|
| M_Exp | Perturbation of input data | 3 hours | - |
| MP_Exp (1) | Perturbation of input data & model parameters | Daily | - |
| MP_Exp (2) | Perturbation of input data & model parameters | Daily | Sensitivity to SWE measurement frequency |
| MPP_Exp | Perturbation of input data & model parameters | Daily | Additional snow density model |
| nP_Exp | Perturbation of input data & model parameters | Daily | Sensitivity to the particles number |

**Table 5: List of multivariate DA experiments.**

| | Surface temperature [°C] | | | SWE [mm] | | | Snow depth [m] | | |
|---|---|---|---|---|---|---|---|---|---|
| | CDP | WFJ | TGN | CDP | WFJ | TGN | CDP | WFJ | TGN |
| M_Exp | 0,097 | 0,128 | 0,112 | -1,040 | 0,007 | 0,365 | 0,651 | 0,764 | 0,557 |
| MP_Exp (1) | 0,108 | 0,062 | 0,111 | 0,746 | 0,007 | 0,583 | 0,813 | 0,833 | 0,694 |
| MPP_Exp | - | 0,050 | 0,092 | - | 0,467 | 0,639 | - | 0,701 | 0,565 |

**Table 6: Comparison of DA performance under different configurations: values of CRPSS score.**

|          | Surface temperature [°C] | | | SWE [mm] | | | Snow depth [m] | | |
|----------|-------|-------|-------|-------|-------|-------|-------|-------|-------|
|          | CDP   | WFJ   | TGN   | CDP   | WFJ   | TGN   | CDP   | WFJ   | TGN   |
| OL-Ens   | 0,369 | 0,774 | 0,591 | 0,893 | 0,864 | 0,361 | 0,182 | 0,855 | 0,798 |
| 100-Ens  | 0,494 | 0,744 | 0,616 | 0,995 | 0,937 | 0,450 | 0,937 | 0,959 | 0,812 |
| 200-Ens  | 0,484 | 0,741 | 0,636 | 0,991 | 0,959 | 0,466 | 0,925 | 0,990 | 0,814 |
| 500-Ens  | 0,485 | 0,743 | 0,617 | 0,993 | 0,962 | 0,537 | 0,933 | 0,968 | 0,819 |

**Table 7: nP-Exp - KGE values of the ensemble mean simulations as a function of the particles number.**