# Peer review of "A Particle Filter scheme for multivariate data assimilation into a point-scale snowpack model in Alpine environment"

_The Cryosphere, 2017_

## Referee Comment (RC1) · M. Lafaysse (Referee) · 6 Feb 2018

General remarks

- I think that this manuscript is a very significant contribution in the field of data assilimation for snowpack modelling. The originality of this paper comes from the multivariate assimilation in the context of the particle filter algorithm. Another added value is the multi-sites application whereas recent applications of the particle filter in snowpack modelling were only focused on one specific site. The multivariate assimilation exhibits some promising advantages but also some discrepancies and challenges which have to be accounted for in the development of such systems. The paper gives a very in-

teresting overview of these positive and negative effects, and their links with the model structure and with the frequency of available observations.

- The introduction gives a very good overview of the position of this work among state-of-the-art methods.

- I have the feeling that the structure of the paper could be a bit improved before publication in two ways:

1/ First, the results section is a bit too long because it includes some details of the methodology itself which should be described in section 2. This is especially the case because the section already includes both results description and discussion. In particular, the beginning of sections 3.1.1, 3.1.2, 3.2.1, 3.2.2 and 3.3.1 introduce many methodological elements which could be detailed in section 2. I would suggest a paragraph 2.6 describing all the assimilation experiments and their objective. Thus, the presentation of results can become more concise.

2/ Then, the authors should better emphasize the lessons of their work for current and future developments of data assimilation in snowpack modelling systems, in a more general point of view than their particular study. This could be done either by introducing a dedicated discussion section either by adding complementary informations and perspectives in the conclusion. For example, the challenge of spatialization at larger scales should be mentioned because it will be a major issue for hydrological modelling. Then, can the authors give general recommendations for the implementation of data assimilation algorithms further than their particular case? From their results, do they recommend to always include parameter perturbations? Do they recommend to include parameter perturbations this way or to test other methods? Do they recommend to apply restrictions in terms of availability of observations to decide to assimilate a given variable? Do they recommend a minimal model structure to decide to assimilate some specific variables ?

Major issues

- My main concern is the fact that the skill of data assimilation is assessed by the comparison of deterministic scores between ensemble simulations including data assimilation and the deterministic reference simulation which is forced by in-situ meteorological measurement. However, in the real world, the quality of the meteorological forcing will be much lower that the quality of the forcing at the three stations of Col de Porte, Weissfluhjoch and Torgnon. Therefore, it makes sense to use perturbations which are not really representative of the uncertainty of these meteorological dataset but more typical of common meteorological errors. Although it is not clearly said in the paper (section 2.4.1), this is what is done here because the error statistics of Charrois et al, 2016 and Magnusson et al, 2017 come from a comparison between a meteorological analysis and in-situ observations. These errors do not represent the observation error, they represent the meteorological analysis error. As a consequence, data assimilation is expected to reduce the meteorological error introduced in the forcing. But it is very demanding to expect from data assimilation to come back to results of the same quality as simulations forced by in-situ measurements when perturbations higher than the observation uncertainty are introduced. There are several options to solve this issue:

Option 1) using lower perturbations consistent with the meteorological forcing. The main limitation will be a low spatial transferability of the results as very few stations provide this quality of meteorological data.

Option 2) using a meteorological forcing of lower quality more consistent with the perturbations. This option would require to run again all simulations.

Option 3) changing the evaluation metrics to provide a comparison of skill between 2 ensembles, the first one with the perturbations but without assimilation and the second one with assimilation. This option does not imply to change the simulation runs, it only requires to compute new evaluation metrics. Therefore, I would recommend this option for this work. The easiest way will be to keep the same metrics but to apply them to the ensemble without assimilation instead of the reference run without perturbation. Thus, the blue points in Fig. 4, 6, 10 will be replaced by a boxplot which can be compared with

the red boxplot (ensemble with assimilation). Note that it would also be possible to use ensemble metrics instead of deterministic metrics. For example you could compute the Continuous Ranked Probability Score (CRPS) of the ensembles with and without assimilation.

- The second major issue is the fact that the scores are presented in a very high number of subplots (Fig. 4, 6, 10) which are very small. The comparison of the different experiments is difficult with these figures due to the lack of more synthetic metrics allowing a quicker comparison of the experiments. It is probably interesting to see the interannual variability of the scores for one example but I do not think that this is necessary for all scores, sites, and experiments. It is impossible to analyze in details all the scores provided in these 3 figures. Page 16, line 11, clearly the authors do not need all the metrics of Figure 10 for such a general conclusion! I think the authors should try to present multi-year scores in a synthetic table allowing a quick and representative overview of the model skill for the different experiments.

Other remarks

Page 1 line 24: It would be useful to also mention that snow models are based on uncertain parameterizations and parameters (Essery et al, 2013; Lafaysse et al, 2017). Thus, it would become more natural to introduce further the perturbations of model parameters.

Page 1 line 28: It is not obvious that there is a link between the complexity and the skill of the data assimilation algorithm.

Page 1 line 29: "they allow to process" → they allow taking benefit from

Page 2 line 4: snow models (plural)

Page 2 lines 5-8: Optimal interpolation also allows accounting for observation uncertainty.

Page 2 line 15: EnKF can also be based on ensembles obtained from other methods

than the Monte-Carlo approach.

Page 2 lines 21-30: The authors could also add that in the context of more complex models, EnKF is also complicated by the need to compute averages of the snowpack profiles. This can be a challenge for the models based on a lagrangian discretization with a variable number of snowpack layers.

Page 2 line 33: "the full prior density" → coming from the ensemble

Page 3 lines 18-20: Please add "at the local scale" because these conclusions might not be true in spatialized simulations.

Page 4, lines 9-10: I do not agree that instrumental biases are representative of observation uncertainties. Even on these well-maintained sites, environmental errors are the prevailing source of uncertainty. Therefore, the instrumental accuracy provided by manufacturers does not provide a good assessment of observation error. For example, the radiation sensors are generally more affected by environmental issues (hoar or snow on the sensor) than for instrumental accuracy. Similarly, precipitation measurement is mainly affected by undercatch in case of wind.

Page 4, line 13: "all the requirements" → to force and evaluate a snow model

Page 4, lines 23-28; page 5, lines 2-7: I think that it is not necessary to provide so many details about the available observations at Col de Porte and Weissfluhjoch. The observations which are not used in this paper (temperature profiles, ground temperatures, liquid water content, runoff, etc.) do not need to be described.

Page 5, line 30: Can you detail what represent the 2 distinct layers? I assume that there is a surface layer? Does it have a fixed depth?

Section 2.2 Can you explain how the energy balance is computed without the availability of a longwave radiation forcing?

Page 6 line 27: model input vector → meteorological input vector

[Figure]

Page 6 line 32: why do you prefer here the word Âńnoise¿ to Âńerror¿? I think it would be more accurate to talk about observation error.

Page 7 line 4: missing space after t-1

Page 7 line 12: This statement could be more general. Indeed, as mentioned before, the Monte Carlo sampling is not the only method to build an ensemble.

Page 7 lines 17-18: I followed the formalism until here but I do not fully understand the sentence ÂńParticles are drawn from a known proposal distribution according to the Sequential Importance Sampling approach¿. Can you clarify this part so that it can be understood without reading the references associated with the SIS approach?

Page 8 line 6: I think that the reference to Fig. 2 in the text does not take all the benefit of this figure to clarify the methodology. I would suggest to refer separately to the different subplots in the text to be more illustrative. Can you also comment the reasons which explain the slight differencies between Fig 2b (weights) and 2d (number of resampled particles)?

Page 8 equation 10: Can you explain by words the practical implication of this equation?

Page 8 line 23: Âńadditive stochastic noise¿ Can you detail the process applied for precipitation? I assume it is probably not possible to apply directly an additive noise in that case? Is there a different treatment between occurrency and intensity?

Page 9 line 1-2: I agree with this remark. However, following my first major remark, the perturbations used in this study are not representative of the error of the in-situ measurements at Col de Porte.

Page 9 lines 4-6: The perturbation of model parameters is introduced through a very "mechanical" point of view for the data assimilation algorithm. I think it would be useful to remind that errors exist in the snow model itself and that it is natural that perturbations of the meteorological inputs are not sufficient to cover all uncertainty.

From page 9 line 29 to page 10 line 3: Are the authors aware that weekly measurement of bulk density are available at Col de Porte at 3 different places in the plot? These data should be preferred for data assimilation than a computation from SWE and snow depth. Indeed, very unrealistic values are obtained with such a computation because the spatial variability in the plot is responsible for a different accumulation between both sensors.

Page 10 Line 14 Can you provide the time step and the hour used for the surface temperature?

Page 10 Line 16 "snowless periods are neglected" How do you define the snow free periods? Is it only based on observations? This choice can lead to eliminate some data for which some particles do have snow and to include data for which some particles do not have snow. This is a usual issue in the evaluations of a snowpack model so please be accurrate on that point.

Page 10 line 28: "spurious trends" → unexpected biases

Page 10 lines 25-28: The goal of section 3.1.1 should also be to check if the perturbations are able to realistically depict the uncertainty of snow simulations.

Page 11 line 5: It would be interesting to notice that despite unbiased perturbations, the control run is not identical to the ensemble mean.

Page 11 line 6: A more comprehensive assessment of the fact the control run is included in the ensemble spread would be to use Talagrand rank histograms over the whole period to ckeck that the control run has a random position in the ensemble. Note that it would be even more informative to check if the observation is included in the ensemble spread with a random position. This would be very useful to strengthen the discussion between lines 1-21 of page 12.

Page 12 line 7 "their measures" → the measure of the corresponding variable

Page 13 line 15 "uselessly" Can you be more explicit? Larger computation requirements without a significant improvement of the spread and further of the filter efficiency.

Page 13 lines 18-20: It is unclear if more parameters were also tested in a preliminary sensitivity analysis and which specific metric was used to select the parameters to disturb. Note that the parameters could also be chosen a priori based on previously published sensitivity analyses of other snowpack models.

Page 13 lines 22-23: The albedo parameters could also have an impact of snow mass during the melting season.

Page 13 line 30: Can you provide a better description of Fig 5 in the text? The fact that the spread of viscosity is increasing in the melting period should be noticed. Does it suggest that melting issues in the model are compensated by this parameter?

Page 15 line 28: Can you give more details about the new density function and how it differs from the original relationship between SWE and snow depth in your model?

---

## Referee Comment (RC2) · Anonymous Referee #2 · 9 Mar 2018

General comments

In this study, the authors test different particle filer setups for jointly assimilating a set of snowpack variables, such as snow depth, SWE and snow surface temperature. The study is a valuable contribution to previous studies, which have assessed the performance of the particle filter for the assimilation of only one snowpack variable in most cases. However, the study needs some improvements before final publication. Four important issue are:

- The authors only use 100 particles when testing the performance of the filter. In some situations, such a low number of particles might give good filter performance.

[Figure]

However, in the case of multivariate assimilation of several variables, more particles may be needed. Therefore, I would urge the authors to test the sensitivity of the filter performance by varying the number of particles. The authors should also present results showing the effective sample size after each update in order to test whether the number of particles is sufficient.

- I could not find any information about the uncertainty of the different measurement. The specification of the observation uncertainties is critically important for the filter behavior and should be reported.

- Some of the figures contain too much information, foremost figure 4, 6 and 10. The large number of results shown in these figures makes it hard to see which filter setup performs best. This is further complicated by the different axes limits used in the figures (see for example the performance metric NER for CPD and SWE in figure 4, 6 and 10). Overall, I think the presentation of the results would improve by removing some of the performance metrics. The conclusions from this study may also be clearer if the authors could summarize their results in fewer graphs.

- First, the result sections contain more discussions about methods, rather than presentations of their results and quantitative comparisons between them. Second, the authors often states that one filter setup performs better than another setup. However, how large those improvements are is not presented in numbers between the setups. It is therefore very hard to judge whether the simulation results actually improved.

Specific comments

Abstract: I think the abstract lacks clear "take home messages". What are the most important results and conclusions obtained in this study?

Page 2, Lines 1-8: It is also possible to include observation uncertainties using the optimal interpolation scheme.

Page 2, Lines 11-15: I think the Enkf was not invented "with the aim of overcoming the

inaccuracy of the linearization procedure", but to avoid the need for linearization of the system equations, which in many cases is impossible or simply unfeasible.

Page 2, Lines 24-32: Please state the study goals in more detail using, for example, research questions or hypothesis. Perhaps remove the summary part stretching from line 27 to 32.

Section 2.1: The Torgnon site description includes information about the measurement equipment, whereas the other site descriptions lack this information. I think it would be better to present the same amount of information for each of the field site. If including information about the measurement equipment for all field sites, perhaps better add a table to the paper with this information. Furthermore, the Torgnon site description includes some numbers about climatic conditions. Such information should be included for the two other field sites as well.

Page 7, Lines 26-28: I do not understand this part of the sentence: "a resampling procedure is frequently introduced to restore the sample variety through a Markov chain chaotic Monte Carlo". What is a "Markov chain chaotic Monte Carlo"?

Equation 8: The effective sample size should be calculated using the square root of the weights.

Section 2.3.2: What uncertainty was used for the different observations? This information is essential and must be included in the manuscript.

Page 8, Lines 19-20: Why was not longwave radiation perturbed?

Page 9, Line 14: I do not understand this sentence: "Therefore, tuning parameters are properly set to guarantee a significant variance of the parameters distribution."

Section 2.4.2: How are the parameter values perturbed? By additive or multiplicative noise?

Page 10, Lines 10-12: Perhaps remove: "The SWE is one of the most relevant snowrelated quantities from a hydrological point of view, since its accuracy in estimate strongly impacts discharge simulations".

Equation 11 and 12: These two equations are probably not needed since the two metrics are very common.

Page 12, Lines 1-21: In this part of the manuscript, I think the authors are mainly discussing filter degeneracy, which is a well-known problem for these kind of applications, in particular when the dimensions of the observation space is high. Please shorten this general discussion by citing relevant literature (e.g. Ades et al., 2013), and provide results more specific to the actual study. Whether such a degeneracy occurs can be assessed by either calculating the efficient sample size, or qualitatively by plotting the time series of the particle spread after assimilation. It would probably be good to include one or both of those analyses to the result sections.

Ades, M. and P. J. Van Leeuwen, 2013: An exploration of the equivalent weights particle filter. Quartely Journal of Meteorology, 139, 820-840.

Page 14, Line 8: In many places throughout the manuscript the authors refers to "parameters resampling" or similar terminology. I do not understand this terminology, and I am not sure it is a correct since particles are being replicated or terminated in the resampling step, and parameter values are only affected indirectly. Please consider rephrasing.

Page 14, Lines 8-10: Please add some quantitative measures on how large this improvement actually is.

Page 14, Lines 10-11: How are the model parameters better estimated? What are the best values of the parameters?

Page 15, Lines 8-20, including Figure 8: The spread between the particles in the figures seems very small, indicating sample impoverishment. I suspect that the number of particles is not high enough for these kind of experiments, or that a more appropriate

filter technique for high dimensional problems should be used requiring fewer particles. Please analyze whether sample impoverishment is occurring or not, and provide appropriate results from such an analysis in the manuscript.

Page 16, Lines 9-11: Provide quantitative results on how much the simulation improves.

Page 16, Lines 9-20: What observation uncertainty was assigned to the "proxy information of snow mass-related variables"?

Page 16, Line 30: I think it should be "perturbations of parameters" instead of "parameter resampling" as mentioned above.

Conclusions: The conclusions mainly lists problems with the SIR-PF for the current application. In addition, I would like to know the answer to questions like these: What filter setup worked best for the current application? Does the filter work better for sites with low (CDP) or high (WFJ) snow amounts? What assimilation frequency worked best? Such information is currently missing in the conclusions.

Technical corrections

Page 1, Line 10: Perhaps remove "multivariate Sequential Importance Resampling".

Page 4, Line 34: Probably wrong reference: Wever, 2015?

Page 10, Lines 15-21: All variables are not explained (e.g. Exp, Obs). It is pretty clear what they mean, but I think for completeness they should be described.

Page 12, Line 14: Change to: "Firstly, it is intended to properly identify the parameters affecting the model simulations most" or something better.

---

## Author Comment (AC1) · 17 Apr 2018

**Response to interactive comment from Referee #1 (Matthieu Lafaysse):**

Authors responses are shown in blue.

General remarks

- I think that this manuscript is a very significant contribution in the field of data assimilation for snowpack modelling. The originality of this paper comes from the multivariate assimilation in the context of the particle filter algorithm. Another added value is the multi-sites application whereas recent applications of the particle filter in snowpack modelling were only focused on one specific site. The multivariate assimilation exhibits some promising advantages but also some discrepancies and challenges which have to be accounted for in the development of such systems. The paper gives a very interesting overview of these positive and negative effects, and their links with the model structure and with the frequency of available observations.

On behalf of all authors, we thank Matthieu Lafaysse for his detailed and relevant suggestions, which have allowed us to significantly improve our manuscript.

- The introduction gives a very good overview of the position of this work among state of-the-art methods.

- I have the feeling that the structure of the paper could be a bit improved before publication in two ways:

1/ First, the results section is a bit too long because it includes some details of the methodology itself which should be described in section 2. This is especially the case because the section already includes both results description and discussion. In particular, the beginning of sections 3.1.1, 3.1.2, 3.2.1, 3.2.2 and 3.3.1 introduce many methodological elements which could be detailed in section 2. I would suggest a paragraph 2.6 describing all the assimilation experiments and their objective. Thus, the presentation of results can become more concise.

We thank you for this useful remark, which allows us to markedly enhance the readability of our manuscript. We have revised the results section by properly separating the description of methodology from the results discussion. As you suggested, we have introduced a new section, namely Sect. 2.5.2, which presents all the assimilation experiments, listed in the current Table 5. Furthermore, with the aim of making the manuscript more consistent, we propose a further section, namely Sect. 2.5.3, focused on the detailed description of the open loop simulations, that are the reference ensemble simulations, as suggested in your major issues.

In detail, we have moved:

- the beginning of Sect. 3.1.2, p. 11 l. 19-22 in Sect. 2.5.2 to introduce the first experiment [M_Exp], namely the DA simulations with the perturbation of meteorological forcing;

- the beginning of Sect. 3.2, p. 13 l. 8-11 in Sect. 2.5.2 where the second experiment [MP_Exp(1)] is described, namely the DA simulations with the perturbation of meteorological forcing and model parameters;

- Sect. 3.2.1, p. 13 l. 13-28, namely the preliminary analysis of model parameters, in Sect. 2.4.2;

- Sect. 3.2.1, p. 13 l. 29-30 in Sect. 2.5.2 with the aim of improving the consistency of our manuscript;

- Sect. 3.2.2, p. 14 l. 2-6 in Sect. 2.5.2, where the second experiment [MP_Exp(1)] is described, namely the DA simulations with the perturbation of meteorological forcing and model parameters;

- Sect. 3.3.1, p. 15 l. 23-25 in Sect. 3.4, namely the section focused on the fourth experiment [MPP_Exp];

- Sect. 3.3.1, from p. 15 l. 25 to p. 16 l. 7, in Sect. 2.5.2, where the fourth experiment [MPP_Exp] is described, namely the DA simulations with the additional snow density model.

Furthermore, we propose a new numbering of the third Section "Results and Discussion":

2/ Then, the authors should better emphasize the lessons of their work for current and future developments of data assimilation in snowpack modelling systems, in a more general point of view than their particular study. This could be done either by introducing a dedicated discussion section either by adding complementary informations and perspectives in the conclusion. For example, the challenge of spatialization at larger scales should be mentioned because it will be a major issue for hydrological modelling. Then, can the authors give general recommendations for the implementation of data assimilation algorithms further than their particular case? From their results, do they recommend to always include parameter perturbations? Do they recommend to include parameter perturbations this way or to test other methods? Do they recommend to apply restrictions in terms of availability of observations to decide to assimilate a given variable? Do they recommend a minimal model structure to decide to assimilate some specific variables?

We are grateful to the reviewer this remark. Actually we discussed the results focusing on our analysed case studies since it is a first attempt to implement a multivariate PF scheme in the framework of snow modelling. However, as you suggested, we have introduced more general considerations and recommendations, according to our experience. We have stressed the importance of jointly perturbing both the meteorological data and model parameters, especially when dealing with spatialized systems. Furthermore, we have highlighted the potential of using indirect estimates of model state variables with the aim of limiting the system sensitivity to the measurement frequency.

Major issues

- My main concern is the fact that the skill of data assimilation is assessed by the comparison of deterministic scores between ensemble simulations including data assimilation and the deterministic reference simulation which is forced by in-situ meteorological measurement. However, in the real world, the quality of the meteorological forcing will be much lower that the quality of the forcing at the three stations of Col de Porte, Weissfluhjoch and Torgnon. Therefore, it makes sense to use perturbations which are not really representative of the uncertainty of these meteorological dataset but more typical of common meteorological errors. Although it is not clearly said in the paper (section 2.4.1), this is what is done here because the error statistics of Charrois et al, 2016 and Magnusson et al, 2017 come from a comparison between a meteorological analysis and in-situ observations. These errors do not represent the observation error, they represent the meteorological analysis error. As a consequence, data assimilation is expected to reduce the meteorological error introduced in the forcing. But it is very demanding to expect from data assimilation to come back to results of the same quality as simulations forced by in-situ measurements when perturbations higher than the observation uncertainty are introduced. There are several options to solve this issue: Option 1) using lower perturbations consistent with the meteorological forcing. The main limitation will be a low spatial transferability of the results as very few stations provide this quality of meteorological data. Option 2) using a meteorological forcing of lower quality more consistent with the perturbations. This option would require to run again all simulations. Option 3) changing the evaluation metrics to provide a comparison of skill between 2 ensembles, the first one with the perturbations but without assimilation and the second one with assimilation. This option does not imply to change the simulation runs, it only requires to compute new evaluation metrics. Therefore, I would recommend this option for this work. The easiest

way will be to keep the same metrics but to apply them to the ensemble without assimilation instead of the reference run without perturbation. Thus, the blue points in Fig. 4, 6, 10 will be replaced by a boxplot which can be compared with the red boxplot (ensemble with assimilation). Note that it would also be possible to use ensemble metrics instead of deterministic metrics. For example you could compute the Continuous Ranked Probability Score (CRPS) of the ensembles with and without assimilation.

This remark is definitively of key importance and we would like to thank the reviewer for pointing out this issue. As suggested, we have chosen the Option n°3, namely we have considered the probabilistic open loop run as the control one. Therefore, we have compared the ensemble simulations resulting from each experiment with the ensemble open loop simulations. With the aim of improving the comparison among the experiments, we are also proposing to replace the previous 4 statistical indices, namely Correlation coefficient, RMSE, Efficiency and Net Error Reduction, with only 2 evaluation metrics. The first one is the Kling-Gupta Efficiency (KGE) coefficient, a deterministic metrics allowing to jointly take account of the correlation coefficient, an estimate of the relative variability between simulated and observed quantities, and a measure of the overall bias. We have replaced Figures 4, 6, and 10 with the current Figure 7, which strictly compares the multi-year KGE values resulting from all the experiments. The second newly-introduced evaluation metrics is an ensemble-based probabilistic score, namely the Continuous Ranked Probability Skill Score (CRPSS), whose values are listed in an overview table ensuring a quick comparison among the different DA configurations (current Table 6).

Furthermore, with the aim of improving the clarity of our manuscript, we have also modified Sect. 2.4.1, p. 9, l. 5-7:

**"Even though this approach ensures to take account of the actual meteorological errors affecting the quality of the model predictions, the main limitation of this procedure is the lack of correlations among the perturbed forcing variables, which does not ensure their physical consistency (Charrois et al., 2016)."**

- The second major issue is the fact that the scores are presented in a very high number of subplots (Fig. 4, 6, 10) which are very small. The comparison of the different experiments is difficult with these figures due to the lack of more synthetic metrics allowing a quicker comparison of the experiments. It is probably interesting to see the interannual variability of the scores for one example but I do not think that this is necessary for all scores, sites, and experiments. It is impossible to analyze in details all the scores provided in these 3 figures. Page 16, line 11, clearly the authors do not need all the metrics of Figure 10 for such a general conclusion! I think the authors should try to present multi-year scores in a synthetic table allowing a quick and representative overview of the model skill for the different experiments.

As previously explained, we have introduced more synthetic evaluation metrics, namely the KGE and CRPSS scores, to allow a quicker comparison among the experiments results. Actually, we agree that the interannual variability of the scores is not of significant relevance, since the main goal is to assess the overall performance of each multivariate DA configuration. Therefore, in place of Figures 4, 6, and 10, we are proposing the current Figure 7, which shows multi-year KGE scores. Moreover, we have listed the resulting CRPSS indices in the current Table 6.

Other remarks

Page 1 line 24: It would be useful to also mention that snow models are based on uncertain parameterizations and parameters (Essery et al, 2013; Lafaysse et al, 2017). Thus, it would become more natural to introduce further the perturbations of model parameters.

In this sentence we list some of the main real-world phenomena and causes which make it difficult to model the snowpack dynamics. Here we are not including modelling issues. Therefore, we are proposing to

introduce the parameters uncertainty directly at the beginning of the current Section 2.4.2, namely the Section focused on the parameters perturbation.

Page 1 line 28: It is not obvious that there is a link between the complexity and the skill of the data assimilation algorithm.

We agree with this remark. We have revised this sentence.

Page 1 line 29: "they allow to process" → they allow taking benefit from

We have accordingly revised this sentence.

Page 2 line 4: snow models (plural)

We have accordingly revised the text.

Page 2 lines 5-8: Optimal interpolation also allows accounting for observation uncertainty.

We thank you for pointing out this relevant mistake. We have revised this short paragraph.

Page 2 line 15: EnKF can also be based on ensembles obtained from other methods than the Monte-Carlo approach.

We have accordingly revised this sentence.

Page 2 lines 21-30: The authors could also add that in the context of more complex models, EnKF is also complicated by the need to compute averages of the snowpack profiles. This can be a challenge for the models based on a lagrangian discretization with a variable number of snowpack layers.

Thank you for this remark. We have added this further consideration.

Page 2 line 33: "the full prior density" → coming from the ensemble

We have revised this sentence.

Page 3 lines 18-20: Please add "at the local scale" because these conclusions might not be true in spatialized simulations.

We have accordingly revised this sentence.

Page 4, lines 9-10: I do not agree that instrumental biases are representative of observation uncertainties. Even on these well-maintained sites, environmental errors are the prevailing source of uncertainty. Therefore, the instrumental accuracy provided by manufacturers does not provide a good assessment of observation error. For example, the radiation sensors are generally more affected by environmental issues (hoar or snow on the sensor) than for instrumental accuracy. Similarly, precipitation measurement is mainly affected by undercatch in case of wind.

We agree that this statement was improper. Therefore, we have removed this sentence.

Page 4, line 13: "all the requirements" → to force and evaluate a snow model

We have accordingly revised the sentence.

Page 4, lines 23-28; page 5, lines 2-7: I think that it is not necessary to provide so many details about the available observations at Col de Porte and Weissfluhjoch. The observations which are not used in this paper (temperature profiles, ground temperatures, liquid water content, runoff, etc.) do not need to be described.

Thank you for this suggestion. We have removed extra information on observations that are not used in our study.

Page 5, line 30: Can you detail what represent the 2 distinct layers? I assume that there is a surface layer? Does it have a fixed depth?

Yes, there is a surface layer. The thickness of the snow layers can vary and no limit is set for any of them. The snow distribution between the two layers is ruled by the empirical parameterization, which allows maintaining the surface layer thinner than the underlying one. This approach is intended to allow to consider the top layer temperature as an acceptable approximation of the skin temperature, whose measures can thus more efficiently assimilated. We refer to Piazzi et al. (2018, accepted) for the detailed description of the snow modelling scheme.

Section 2.2 Can you explain how the energy balance is computed without the availability of a longwave radiation forcing?

The longwave radiation is not a model input. Both the longwave radiation terms (i.e. incoming and outgoing components) are estimated through the Stephan-Boltzmann law. The outgoing term is calculated as a function of the surface temperature of snow or soil in snowy or snowless conditions, respectively. The incoming component is estimated as a function of the air temperature. While the emissivities of snow and soil are considered as constant model parameters, the air emissivity is time variant and it is evaluated according to both wind speed and air temperature.

Page 6 line 27: model input vector → meteorological input vector

We have accordingly revised the sentence.

Page 6 line 32: why do you prefer here the word "noise" to "error"? I think it would be more accurate to talk about observation error.

We agree. We have replaced "observational noise" with "observation error".

Page 7 line 4: missing space after t-1

We have added the lacking space.

Page 7 line 12: This statement could be more general. Indeed, as mentioned before, the Monte Carlo sampling is not the only method to build an ensemble.

Thank you, we have revised this sentence.

Page 7 lines 17-18: I followed the formalism until here but I do not fully understand the sentence "Particles are drawn from a known proposal distribution according to the ´ Sequential Importance Sampling approach". Can you clarify this part so that it can ˙ be understood without reading the references associated with the SIS approach?

Actually a further binding sentence was lacking. Therefore, we have revised the text by extending the explanation: "**It is noteworthy to consider that the direct sampling of particles from the posterior density is generally difficult, since its distribution is often non-Gaussian. Therefore, particles […]**".

Page 8 line 6: I think that the reference to Fig. 2 in the text does not take all the benefit of this figure to clarify the methodology. I would suggest to refer separately to the different subplots in the text to be more illustrative. Can you also comment the reasons which explain the slight differencies between Fig 2b (weights) and 2d (number of resampled particles)?

We think that the main reason of these slight differences can result from a combined effect depending on both the drawn sample and the shape of the empirical cumulative distribution. We have added the references to each single subplot.

Page 8 equation 10: Can you explain by words the practical implication of this equation?

This equation describes how the particles weights are updated at each assimilation time step, namely by evaluating the value of the likelihood function, which is assumed to be a multi-dimensional Gaussian distribution. The likelihood value of each particle depends on how it is placed with respect to all the available observations.

Page 8 line 23: "additive stochastic noise" Can you detail the process applied for precipitation? I assume it is probably not possible to apply directly an additive noise in that case? Is there a different treatment between occurrence and intensity?

Following the approach proposed by Magnusson et al. (2017), for the precipitation, as well as for wind speed, we assumed an additive stochastic noise having a lognormal distribution, and this is now specified in the text. We perturbed only the precipitation intensity.

Page 9 line 1-2: I agree with this remark. However, following my first major remark, the perturbations used in this study are not representative of the error of the in-situ measurements at Col de Porte.

Thank you for this comment. According to your main remark, in Sect. 2.4.1 we have better specified that the perturbations are not representative of the error of ground-based measurements, rather of the common meteorological analysis error.

Page 9 lines 4-6: The perturbation of model parameters is introduced through a very "mechanical" point of view for the data assimilation algorithm. I think it would be useful to remind that errors exist in the snow model itself and that it is natural that perturbations of the meteorological inputs are not sufficient to cover all uncertainty.

We agree with this remark. Actually we introduced and discussed the perturbation of model parameters only with regard to its potential within the DA framework. Firstly, we are proposing to modify the title of the Section 2.4.2, "**Perturbation of model parameters**". Secondly, we have added an introduction sentence at the beginning of this Section.

From page 9 line 29 to page 10 line 3: Are the authors aware that weekly measurement of bulk density are available at Col de Porte at 3 different places in the plot? These data should be preferred for data assimilation than a computation from SWE and snow depth. Indeed, very unrealistic values are obtained with such a computation because the spatial variability in the plot is responsible for a different accumulation between both sensors.

Thank you for this interesting remark. Actually we knew that weekly measurements of bulk density are available at the Col de Porte. We have chosen to use indirectly derived estimates of snow density since they can be evaluated with a daily frequency, which is an interesting benefit to test the system sensitivity to difference in the measurements frequency, with respect to the other sites. We have taken account of the higher uncertainty of the density estimates with respect to the other variables. Before using these snow density estimates, we have qualitatively assessed the consistency of their values, neglecting those deemed unreliable.

Page 10 Line 14 Can you provide the time step and the hour used for the surface temperature?

The surface temperature is updated every 15 minutes, according to the model integration time step. The observations are assimilated every 24 hours, at 11 a.m.

Page 10 Line 16 "snowless periods are neglected" How do you define the snow free periods? Is it only based on observations? This choice can lead to eliminate some data for which some particles do have snow and to

include data for which some particles do not have snow. This is a usual issue in the evaluations of a snowpack model so please be accurrate on that point.

We have estimated the melt-out date of each winter season according to the observations. For each station, the snowless periods start after the last melt-out date evaluated over the whole dataset, with the aim of being conservative as much as possible.

Page 10 line 28: "spurious trends" → unexpected biases

We have accordingly revised the text (current Sect. 2.5.3).

Page 10 lines 25-28: The goal of section 3.1.1 should also be to check if the perturbations are able to realistically depict the uncertainty of snow simulations.

We have added this further consideration (current Sect. 2.5.3).

Page 11 line 5: It would be interesting to notice that despite unbiased perturbations, the control run is not identical to the ensemble mean.

We have added this further consideration (current Sect. 2.5.3).

Page 11 line 6: A more comprehensive assessment of the fact the control run is included in the ensemble spread would be to use Talagrand rank histograms over the whole period to ckeck that the control run has a random position in the ensemble. Note that it would be even more informative to check if the observation is included in the ensemble spread with a random position. This would be very useful to strengthen the discussion between lines 1-21 of page 12.

As you suggested, we have analysed the Talagrand rank histograms to assess whether both observations and the deterministic control simulations are included within the ensemble spread. We have introduced the current Figure 6 showing the Talagrand histogram of the SWE ensemble open loop simulations throughout the overall CDP dataset.

Page 12 line 7 "their measures" → the measure of the corresponding variable

We have accordingly revised this sentence.

Page 13 line 15 "uselessly" Can you be more explicit? Larger computation requirements without a significant improvement of the spread and further of the filter efficiency.

We have better specified this sentence.

Page 13 lines 18-20: It is unclear if more parameters were also tested in a preliminary sensitivity analysis and which specific metric was used to select the parameters to disturb. Note that the parameters could also be chosen a priori based on previously published sensitivity analyses of other snowpack models.

The preliminary sensitivity analysis involved also other model parameters. We have selected only those a significant impact on the simulations. According to the approach described in Piazzi et al. (2018, accepted), we use the KGE coefficient as evaluation metric. We have added the reference in the text.

Page 13 lines 22-23: The albedo parameters could also have an impact of snow mass during the melting season.

Thank you, actually we omitted this information. We have added this further consideration.

Page 13 line 30: Can you provide a better description of Fig 5 in the text? The fact that the spread of viscosity is increasing in the melting period should be noticed. Does it suggest that melting issues in the model are compensated by this parameter?

Yes, the gradual increase of the ensemble spread (current Figure 3) can definitively suggest an offsetting effect throughout the melting period. Thank you for this interesting remark. We have added these considerations in the text (current Sect. 2.4.2).

Page 15 line 28: Can you give more details about the new density function and how it differs from the original relationship between SWE and snow depth in your model?

According to the approach proposed by Jonas et al. (2009), the snow density is estimated through an empirical parameterization relying on the reported 4 main factors. The authors defined through a linear regression the two coefficients [b, a] best fitting an extended observational dataset of snow depths and snow densities. Therefore, the snow density ($\rho_{estim}$) is evaluated as [$\rho_{estim} = a \cdot SD_{obs} + b$], where $SD_{obs}$ is the observed snow depth. The resulting SWE estimate SWE estimation ($SWE_{estim}$) is retrieved as [$SWE_{estim} = SD_{obs} \cdot \rho_{estim}$].

---

## Author Comment (AC2) · 17 Apr 2018

**Response to interactive comment from Anonymous Referee #2**

Authors responses are shown in blue.

General comments

In this study, the authors test different particle filer setups for jointly assimilating a set of snowpack variables, such as snow depth, SWE and snow surface temperature. The study is a valuable contribution to previous studies, which have assessed the performance of the particle filter for the assimilation of only one snowpack variable in most cases. However, the study needs some improvements before final publication.

On behalf of all authors, we thank Anonymous Reviewer #2 for his/her detailed and relevant suggestions, which have allowed us to significantly improve our manuscript.

Four important issue are:

- The authors only use 100 particles when testing the performance of the filter. In some situations, such a low number of particles might give good filter performance. However, in the case of multivariate assimilation of several variables, more particles may be needed. Therefore, I would urge the authors to test the sensitivity of the filter performance by varying the number of particles. The authors should also present results showing the effective sample size after each update in order to test whether the number of particles is sufficient.

We would like to thank the reviewer for this remark of key importance. Actually, we did not test the system sensitivity to the ensemble size, even though in a multivariate DA application this critical issue need to be addressed. Therefore, as suggested, we have performed a further experiment (called nP_Exp, current Sect. 3.5) with the aim of assessing the effective ensemble size, the ensemble spread (current Figure 11) and the performance of the multivariate DA scheme (current Table 7) as the particles number increases: 100-, 200-, and 500-particles. This experiment considers a sample of one randomly-chosen winter season for each analysed experimental site. As shown and explained in the manuscript, the results generally do not show a significant system sensitivity to the ensemble size.

- I could not find any information about the uncertainty of the different measurement. The specification of the observation uncertainties is critically important for the filter behavior and should be reported.

We have considered the following observational uncertainties: 2°C for the surface temperature; 10 mm for the SWE; 0.15 for the surface albedo; 0.05 m for the snow depth; 50 kg/m$^3$ for the snow density. As required, we have reported this information in the current Table 4.

- Some of the figures contain too much information, foremost figure 4, 6 and 10. The large number of results shown in these figures makes it hard to see which filter setup performs best. This is further complicated by the different axes limits used in the figures (see for example the performance metric NER for CPD and SWE in figure 4, 6 and 10). Overall, I think the presentation of the results would improve by removing some of the performance metrics. The conclusions from this study may also be clearer if the authors could summarize their results in fewer graphs.

Thank you for this useful remark. We definitely agree that the comparison among the multivariate DA configuration was not clear and quite hard to assess. With the aim of ensuring a more concise and effective presentation of the results, we are proposing to replace the previous 4 statistical indices, namely Correlation coefficient, RMSE, Efficiency and Net Error Reduction, with only 2 evaluation metrics. The first one is the Kling-Gupta Efficiency (KGE) coefficient, a deterministic metrics allowing to jointly take account of the correlation coefficient, an estimate of the relative variability between simulated and observed quantities, and a measure of the overall bias. We have replaced Figures 4, 6, and 10 with the current Figure 7, which strictly compares the multi-year KGE values resulting from all the experiments. The second newly-introduced evaluation metrics is an ensemble-based probabilistic score, namely the Continuous Ranked Probability Skill

Score (CRPSS), whose values are listed in an overview table ensuring a quick comparison among the different DA configurations (current Table 6).

- First, the result sections contain more discussions about methods, rather than presentations of their results and quantitative comparisons between them.

This remark has allowed us to significantly improve the readability of our manuscript. We have revised the results section by properly separating the description of methodology from the results discussion. We have introduced a new section, namely Sect. 2.5.2, which presents all the assimilation experiments, also listed in the current Table 5. Furthermore, with the aim of make the manuscript more consistent, we are proposing a further section, namely Sect. 2.5.3, focused on the detailed description of the control open loop simulations (ex-Sect. 3.1.1).

In detail, we have moved:

- the beginning of Sect. 3.1.2, p. 11 l. 19-22 in Sect. 2.5.2 to introduce the first experiment [M_Exp], namely the DA simulations with the perturbation of meteorological forcing;

- the beginning of Sect. 3.2, p. 13 l. 8-11 in Sect. 2.5.2 where the second experiment [MP_Exp(1)] is described, namely the DA simulations with the perturbation of meteorological forcing and model parameters;

- Sect. 3.2.1, p. 13 l. 13-28, namely the preliminary analysis of model parameters, in Sect. 2.4.2;

- Sect. 3.2.1, p. 13 l. 29-30 in Sect. 2.5.2 with the aim of improving the consistency of our manuscript;

- Sect. 3.2.2, p. 14 l. 2-6 in Sect. 2.5.2, where the second experiment [MP_Exp(1)] is described, namely the DA simulations with the perturbation of meteorological forcing and model parameters;

- Sect. 3.3.1, p. 15 l. 23-25 in Sect. 3.4, namely the section focused on the fourth experiment [MPP_Exp];

- Sect. 3.3.1, from p. 15 l. 25 to p. 16 l. 7, in Sect. 2.5.2, where the fourth experiment [MPP_Exp] is described, namely the DA simulations with the additional snow density model.

Furthermore, we propose a new numbering of the third Section "Results and Discussion":

- Section 3.1: **Multivariate DA simulations with perturbed meteorological input data**

- Section 3.2: **Multivariate DA simulations with perturbed model parameters**

- Section 3.3: **Sensitivity analysis of the multivariate DA scheme to the SWE measurement frequency**

- Section 3.4: **Multivariate DA simulations with proxy information of snow mass-related variables**

- Section 3.5: **Sensitivity analysis of the multivariate DA scheme to the ensemble size**

Second, the authors often states that one filter setup performs better than another setup. However, how large those improvements are is not presented in numbers between the setups. It is therefore very hard to judge whether the simulation results actually improved.

We have substantially revised the presentation of the experimental results in Section 3, with the aim of improving and making easier the quantitative comparison among the different multivariate DA configurations, through fewer and more comprehensive evaluation metrics (current Figure 7 and Table 6).

Specific comments

Abstract: I think the abstract lacks clear "take home messages". What are the most important results and conclusions obtained in this study?

Thank you for this remark. Actually, the reference to the main results were completely lacking in the Abstract. Therefore, we have added a brief overview of the most important results with the aim of underlining the key conclusions.

Page 2, Lines 1-8: It is also possible to include observation uncertainties using the optimal interpolation scheme.

We thank you for pointing out this relevant mistake. We have revised this short paragraph.

Page 2, Lines 11-15: I think the Enkf was not invented "with the aim of overcoming the inaccuracy of the linearization procedure", but to avoid the need for linearization of the system equations, which in many cases is impossible or simply unfeasible.

Thank you for this remark. We have accordingly revised the manuscript.

Page 2, Lines 24-32: Please state the study goals in more detail using, for example, research questions or hypothesis. Perhaps remove the summary part stretching from line 27 to 32.

As suggested, we have introduced research questions, which definitively improve the readability of this paragraph. We propose to maintain the brief summary providing a quick overview of the manuscript.

Section 2.1: The Torgnon site description includes information about the measurement equipment, whereas the other site descriptions lack this information. I think it would be better to present the same amount of information for each of the field site. If including information about the measurement equipment for all field sites, perhaps better add a table to the paper with this information. Furthermore, the Torgnon site description includes some numbers about climatic conditions. Such information should be included for the two other field sites as well.

We agree that the experimental sites were not homogeneously described. We would prefer not to go into detail of the measurement equipment of each experimental site, since it is not the focus of this Section, aiming at a more general description. Therefore, we have removed all the information on the instrumental equipment at the Italian site. As required, we have also added information on the climatic conditions at the French and Swiss sites.

Page 7, Lines 26-28: I do not understand this part of the sentence: "a resampling procedure is frequently introduced to restore the sample variety through a Markov chain chaotic Monte Carlo". What is a "Markov chain chaotic Monte Carlo"?

The correct wording should be "Markov Chain Monte Carlo" (MCMC), a widely-used method to probe the posterior probability. We have accordingly updated the text.

Equation 8: The effective sample size should be calculated using the square root of the weights.

Thank you for this remark, there was actually a mistake. We have properly revised the Equation 8.

Section 2.3.2: What uncertainty was used for the different observations? This information is essential and must be included in the manuscript.

As required, we have included this information in Table 4, where the assimilated variables are listed.

Page 8, Lines 19-20: Why was not longwave radiation perturbed?

The longwave radiation is not perturbed since it is not a model input. Indeed, both the longwave radiation terms (i.e. incoming and outgoing components) are estimated through the Stephan-Boltzmann law. The outgoing term is calculated as a function of the surface temperature of snow or soil in snowy or snowless conditions, respectively. The incoming component is estimated as a function of the air temperature. While

the emissivities of snow and soil are considered as constant model parameters, the air emissivity is time variant and it is evaluated according to both wind speed and air temperature.

Page 9, Line 14: I do not understand this sentence: "Therefore, tuning parameters are properly set to guarantee a significant variance of the parameters distribution."

We have used the approach proposed in Moradkhani et al. (2015), where the authors defined a small multiplicative coefficient (tuning parameter) to tune the variance of the random noise used to perturb the model parameters. Since in our study the parameters perturbation has been introduced with the main aim of enlarging the ensemble spread, we have properly defined these tuning coefficients in order to ensure a significant spread of the ensemble of the model parameters by still preserving the physical consistency. We have realised that this was a too specific and technical consideration. Therefore, we have revised the sentence.

Section 2.4.2: How are the parameter values perturbed? By additive or multiplicative noise?

After the resampling procedure, the ensemble of the model parameters is restored by perturbing the parameters through an additive noise. We have introduced this lacking information in the manuscript (Sect. 2.4.2).

Page 10, Lines 10-12: Perhaps remove: "The SWE is one of the most relevant snow-related quantities from a hydrological point of view, since its accuracy in estimate strongly impacts discharge simulations".

We have shortened and modified this sentence.

Equation 11 and 12: These two equations are probably not needed since the two metrics are very common.

As previously explained, these two equations have been removed since we are proposing new evaluation metrics.

Page 12, Lines 1-21: In this part of the manuscript, I think the authors are mainly discussing filter degeneracy, which is a well-known problem for these kind of applications, in particular when the dimensions of the observation space is high. Please shorten this general discussion by citing relevant literature (e.g. Ades et al., 2013), and provide results more specific to the actual study. Whether such a degeneracy occurs can be assessed by either calculating the efficient sample size, or qualitatively by plotting the time series of the particle spread after assimilation. It would probably be good to include one or both of those analyses to the result sections.

Ades, M. and P. J. Van Leeuwen, 2013: An exploration of the equivalent weights particle filter. Quartely Journal of Meteorology, 139, 820-840.

Thank you for this useful remark and the suggested reference. We have shortened this part discussing the filter degeneracy. We have added Figure 8, which compares M_Exp and MP_Exp by showing both the particles spread and the effective filter updating of SWE simulations at an assimilation time step. As regards the efficient sample size, we have restricted this analysis within the assessment of the system sensitivity to the ensemble size ([nP_Exp], current Sect. 3.5).

Page 14, Line 8: In many places throughout the manuscript the authors refers to "parameters resampling" or similar terminology. I do not understand this terminology, and I am not sure it is a correct since particles are being replicated or terminated in the resampling step, and parameter values are only affected indirectly. Please consider rephrasing.

We have used "parameters resampling" with the aim of stressing that the particles are here resampled not only in model state-space, rather in the state-parameter space. However, with the aim of improving the

manuscript readability, in most of cases we have replaced "parameters resampling" with "parameters perturbation".

Page 14, Lines 8-10: Please add some quantitative measures on how large this improvement actually is.

We definitively agree that it was hard to compare the experiments. As previously explained, we have introduced a new ensemble-based evaluation score, namely the CRPSS, to quantitatively assess each DA configuration (Table 6).

Page 14, Lines 10-11: How are the model parameters better estimated? What are the best values of the parameters?

In this sentence we mean that since the particles resampling is here performed in the state-parameters space, it is possible to better take account of the parameters seasonality, rather than assuming constant parameters values.

Page 15, Lines 8-20, including Figure 8: The spread between the particles in the figures seems very small, indicating sample impoverishment. I suspect that the number of particles is not high enough for these kind of experiments, or that a more appropriate filter technique for high dimensional problems should be used requiring fewer particles. Please analyze whether sample impoverishment is occurring or not, and provide appropriate results from such an analysis in the manuscript.

The current Figure 10 (previously Figure 8) shows the time series of the mean of both the snow depth and SWE ensemble simulations. In this Figure the ensemble envelop is not shown. Actually the lack of this information can lead to misleading considerations. Therefore, we have better specified the caption.

Page 16, Lines 9-11: Provide quantitative results on how much the simulation improves.

As previously explained, we have newly introduced the CRPSS score, which allows to more properly assess and compare the different multivariate DA configurations (Table 6).

Page 16, Lines 9-20: What observation uncertainty was assigned to the "proxy information of snow mass-related variables"?

We have maintained the same observational uncertainties. Indeed, when increasing the uncertainty of the indirect estimates of the snow mass-related variables (i.e. SWE and snow density) the benefit of assimilating this proxy information is almost nullified due to unbalanced uncertainty values among the assimilated variables.

Page 16, Line 30: I think it should be "perturbations of parameters" instead of "parameter resampling" as mentioned above.

In most cases we have replaced "parameters resampling" with "parameters perturbation".

Conclusions: The conclusions mainly lists problems with the SIR-PF for the current application. In addition, I would like to know the answer to questions like these: What filter setup worked best for the current application? Does the filter work better for sites with low (CDP) or high (WFJ) snow amounts? What assimilation frequency worked best? Such information is currently missing in the conclusions.

Thank you for this useful remark. We have revised the Conclusions to address these missing considerations. We have better pointed out what filter setup works best and what the main limitations are at the analysed sites, according to their local features.

Technical corrections

Page 1, Line 10: Perhaps remove "multivariate Sequential Importance Resampling".

If the reviewer agrees, we would prefer to keep this sentence.

Page 4, Line 34: Probably wrong reference: Wever, 2015?

We are referring to "Wever, N., Schmid, L., Heilig, A., Eisen, O., Fierz, C., and Lehning, M.: Verification of the multi-layer SNOWPACK model with different water transport schemes. The Cryosphere, 9(6), 2271-2293, 2015".

Page 10, Lines 15-21: All variables are not explained (e.g. Exp, Obs). It is pretty clear what they mean, but I think for completeness they should be described.

We have removed all these equations.

Page 12, Line 14: Change to: "Firstly, it is intended to properly identify the parameters affecting the model simulations most" or something better.

We have accordingly updated the text "Firstly, it is intended to properly identify the parameters mostly affecting the model simulations".

---

## Author Response (AR2)

**Response to minor revisions from Referee #1 (Matthieu Lafaysse):**

Authors responses are shown in blue.

I would like to thank the authors for their careful answer to all my comments and suggestions. The paper has been significantly improved. The results are much clearer thanks to the better evaluation metrics and the reference to the open-loop ensemble, and the description of the experiments in the methodology significantly improves the paper structure. The main learnings are also better emphasized in the conclusions. As a result, I think that this paper will be a very important contribution for data assimilation in snowpack models. However, I still identify 4 issues which deserve a last minor revision of the manuscript before publication:

On behalf of all authors, we thank Matthieu Lafaysse for the value he found in our research as well as for his further relevant and helpful suggestions.

1) The new description of the openloop ensemble behaviour is extremely useful (now in Section 2.5.3). However, I am not sure whether the results (from page 12 line 23 to page 13 line 10) are well positioned in the manuscript as it would be more common to present these results at the beginning of section 3.

We thank you for this useful remark allowing to enhance the readability of our manuscript. We have moved the results of Section 2.5.3 at the beginning of Section 3, as Sect. 3.1. The numbering of the following sections has been accordingly updated.

2) In the same part, I find the rank histograms very useful. The right one relative to the rank of the deterministic member is well described in the text but unfortunately the left one relative to the rank of the observation is not described. I think a short description will be very useful. Indeed this rank histogram is typical of an underdispersive system and it suggests that the perturbations of the meteorological input will not be sufficient either because they are lower than the real uncertainty either because perturbations of model parameters are also required.

We have better described the histogram relative to the rank of observations, according to your valuable remarks (current Sect. 3.1).

3) Several answers to my questions improve the understanding of this work but unfortunately some of them were not included in the manuscript whereas they would be very useful. This is the case about:

We have introduced in our manuscript all the information reported in the list.

- the 2 layers characteristics because it helps understanding what is the simulated surface temperature;

Snowpack layering has been included in the bullet-pointed list of Sect. 2.2 (page 6).

- the way the incoming longwave radiation is parameterized because it appears as a potential source of uncertainty which is not accounted for in the perturbations applied in this work;

This information has been introduced after listing the meteorological inputs, at the beginning of Sect. 2.2 (page 6).

- the explanation about Equation 10 because it is critical for the reader to understand what the authors say in their answer;

We have better explained how the particles weights are updated at each assimilation time step (page 8, right after Eq. 10).

- the fact that precipitation intensity is not perturbed;

We have specified that only the precipitation intensity is perturbed at the beginning of Sect. 2.4.1, where the meteorological data are listed (page 9).

- the time of assimilation and evaluation for surface temperature;

As well as the other state variables, the surface temperature is updated every 15 minutes, which is the model integration time step (page 9). As explained in Sect. 2.5.4 (page 13), the evaluation metrics are computed by considering all the available observations at their original measurement frequency. We have specified the assimilation time at the end of Sect. 2.5.2 (page 12).

- the definition of the snow free periods.

The definition of the snow free periods is now properly explained in the manuscript (page 13).

4) It should be specified how unique KGE values are obtained in Table 7. Is it the KGE of the ensemble median or the median of KGE for all members?

We have added this missing information in the caption of Table 7, which reports the KGE values of the ensemble mean simulations.

[revised manuscript text omitted]